



# Evolution of structures and hydrothermal alteration in a Palaeoproterozoic metasupracrustal belt: Constraining paired deformation-fluid flow events in a Fe and Cu-Au prospective terrain in northern Sweden

Joel B. H. Andersson[1], Tobias E. Bauer[1], Edward P. Lynch[2]

[1]Division of Geosciences and Environmental Engineering, Luleå University of Technology, SE 971 87 Luleå, Sweden
[2]Department of Mineral Resources, Geological Survey of Sweden, Box 670, SE 751 28, Uppsala, Sweden.

*Correspondence to*: Joel B. H. Andersson (joel.bh.andersson@ltu.se)

**Abstract.** In this field-based study, a ~90 km long Palaeoproterozoic metasupracrustal belt in the northwestern part of the
Norrbotten ore province (northernmost Sweden) has been investigated in order to characterize its various structural components and thus constrain its structural evolution. In addition, hydrothermal mineral associations are described and linked to identified deformation phases. New geological mapping of five key areas (Eustiljåkk, Ekströmsberg, Tjårrojåkka, Kaitum West and Fjällåsen-Allavaara) indicates two major compressional events ($D_1$, $D_2$) that affected the belt whereas each deformation event can be related to specific alteration styles typical for iron oxide-apatite and iron oxide Cu-Au systems. $D_1$
generated a regionally distributed penetrative $S_1$ foliation and oblique reverse shear zones with southwest block up sense-of-shears in response to NE-SW crustal shortening. $D_1$ is associated with regional scapolite $\pm$ albite alteration formed coeval with regional magnetite $\pm$ amphibole alteration and calcite under epidote-amphibolite metamorphism. During $D_2$, folding of $S_1$ generated steeply south-plunging $F_2$-folds in low strain areas whereas most strain was partitioned into pre-existing shear zones resulting in reverse dip-slip reactivation of steep NNW-oriented $D_1$ shear zones and strike-slip dominated movements along
steep E-W-trending shear zones under brittle-ductile conditions. The hydrothermal alteration linked to the $D_2$ deformation phase is more potassic in character and dominated by K-feldspar $\pm$ epidote $\pm$ quartz $\pm$ biotite $\pm$ magnetite $\pm$ sericite $\pm$ sulphides, and calcite. Our results underline the importance of paired structural-alteration approaches at the regional- to belt-scale to understand the temporal-spatial relationship between mineralized systems. Based on the mapping results and microstructural investigations, as well as a review of earlier tectonic models presented for adjacent areas, we suggest a new structural model
for this part of the northern Fennoscandian Shield. Our new structural model harmonizes with earlier petrological/geochemical tectonic models of the northern Norrbotten area and emphasizes the importance of reactivation of early formed structures.



# 1 Introduction

The northern Norrbotten ore province in northern Sweden represents one of Europe's key mining and exploration areas. For example, about 90% of European iron ore is annually produced from two of the world's largest underground iron mines at Kiirunavaara and Malmberget. These world-class deposits (combined current reserves of c. 1051 Mt @ 43.4% Fe; Loussavaara-Kiirunavaara AB, 2017) comprise iron oxide-apatite (IOA) or "Kiruna-type" mineralization, with the former deposit representing the archetypal example (e.g. Geijer, 1910; 1930). The Aitik Cu-Ag-Au deposit also occurs in northern Norrbotten and is one of the largest open-pit copper mines in Europe (current resource of c. 801 Mt @ 0.22 % Cu, 1.3 g/t Ag and 0.15 g/t Au; (New Boliden AB, 2017). Aitik represents an enigmatic porphyry-style deposit with a protracted ore-forming history that is thought to include an overprinting iron oxide-copper-gold (IOCG)-style mineralization event (e.g. Wanhainen et al., 2005; 2012). Beyond the active mines, numerous Fe and Cu ± Au prospects and deposits occur, making the area one of the most prospective in Europe for IOA- and IOCG-style mineralization (e.g. Carlon, 2000; Billström et al., 2010; Martinsson et al., 2016).

In northern Norrbotten (Fig.1), Paleoproterozoic volcanic-sedimentary successions mainly occur within approximately NNW- to NNE-trending metasupracrustal belts (e.g. Bergman et al., 2001). Although volumetrically minor, these belts are significant because their contained lithotypes and structures provide evidence for the long-lived tectonothermal evolution of this sector of the Fennoscandian (Baltic) Shield (Fig. 1). For example, Rhyacian (c. 2.2 - 2.1 Ga) greenstone sequences record a phase of continental rifting, basin formation, mafic-ultramafic magmatism and sedimentation during extensional tectonism (Gustafsson, 1993; Martinsson, 1997; Melezhik and Fallick, 2010; Lynch et al., 2018a). In contrast, younger Orosirian (c. 1.9 – 1.8 Ga) volcanic and sedimentary rocks provide insights into subduction-related magmatism, regional metamorphism and accretionary-collisional processes during the Svecokarelian (or Svecofennian) orogeny (Skiöld et al., 1988; Öhlander et al., 1999; Bergman et al., 2006; Lahtinen et al., 2015). This phase of convergent tectonism promoted basin inversion and juxtaposed supracrustal rocks of differing character and provenance within several discontinuous, curvilinear metasupracrustal domains that are typically bounded by more aerially extensive syn- to late-orogenic intrusions (Pharaoh and Brewer, 1990; Ahl et al., 2001; Bergman et al., 2001; Sarlus et al., 2017; Luth et al., 2018).

Paleoproterozoic metasupracrustal belts in Norrbotten are also significant from a metallogenic perspective as they preferentially host a variety of base and precious metal mineralization, and thus represent key exploration targets (e.g. Carlon, 2000; Martinsson, 2004). In detail, syn-orogenic sequences deposited between c. 1.90 – 1.87 Ga represent a key metalliferous horizon that locally hosts significant IOA- and IOCG-style mineralization (Romer et al., 1994; Edfelt et al., 2005; Smith et al., 2007; Wanhainen et al., 2012; Westhues et al., 2016). Deposits of both types commonly occur within or immediately adjacent to major ductile-brittle deformation zones which tend to traverse and follow the metasupracrustal belts, suggesting these domains and their constituent lithologies strongly influenced strain localization. Norrbotten Paleoproterozoic metasupracrustal rocks also preserve evidence of both regional- and local-scale metasomatic-hydrothermal alteration and fluid-rock interaction (e.g. Romer, 1996; Frietsch et al., 1997; Edfelt et al., 2005; Bernal et al., 2017). This characteristic affirms the efficiency of





these rock sequences and their associated structures to channel and focus the flow of potential metal-bearing fluids through the supracrustal pile (cf. Oliver and Bons, 2001; Cox, 2005).

Previous studies of Svecokarelian-cycle metasupracrustal rocks in northern Norrbotten have included provincial compilations to ascertain lithostratigraphic and petrogenetic insights (e.g. Freitsch, 1984; Pharoh and Pearce, 1984; Forsell,
1987; Perdahl and Freitsch, 1993; Bergman et al., 2001), and local studies constraining the geological, geochemical, geophysical and/or structural character of sequences hosting Fe and Cu ± Au deposits (e.g. Geijer, 1910; 1920; 1930; 1950; Parak, 1975; Edfelt et al., 2006; Sandrin and Elming, 2006; Smith et al., 2007; Wanhainen et al., 2012; Westhues et al., 2016; 2017; Bauer et al., 2018). With a few exceptions (cf. Wright, 1988; Bergman et al., 2001; Grigull et al., 2018, Lynch et al. 2018), regional compilations have lacked accompanying structural information. Local studies with a structural component (e.g.
Debras, 2010; Edfelt et al., 2006; Wanhainen et al., 2012) have generally not considered the broader significance of deposit-proximal structures in reconstructing deformation and/or fluid flow events for individual belts or the wider region. Thus, deformation zone- or belt-scale investigations of Svecokarelian-related metasupracrustal rocks that include a coupled structural-alteration assessment may provide new insights into the number and character of paired deformation-hydrothermal events affecting a particular belt, with the potential to extrapolate broader tectonothermal implications. This approach may
also help identify new geological vectors to IOA and/or IOCG mineralization applicable to underexplored and geographically isolated metasupracrustal domains in Norrbotten, or analogous terranes elsewhere (cf. Corriveau and Mumin, 2010; Corriveau et al., 2016).

In this paper, we present a predominantly field-based investigation of an Orosirian metasupracrustal belt located about 40 km to the west of Kiruna in northwest Norrbotten, northern Sweden (Figs. 1, 2 and 3). The studied sequence, herein referred
to as the Western Supracrustal Belt (cf. Wright, 1988 – see Section 3 below), extends for about 90 km in a NNW-SSE direction and overlaps with the location of a major ductile-brittle deformation zone that hosts several Fe and Cu ± Au occurrences (e.g. Offerberg, 1967; Witschard, 1975; Edfelt et al., 2005; Frietsch, 1974; Wright, 1988; Lynch et al., 2014). New geological mapping of five key domains is used to ascertain the types, geometries, kinematics and interrelationships of various structural components within the belt, and thus constrain its structural evolution. Additionally, a petrographic and paragenetic assessment
of mappable hydrothermal alteration associated with different lithotypes and/or structures is used to constrain the character and number of fluid-flow events within the metasupracrustal rocks, and attempts to link these hydrothermal events to specific phases of deformation. Overall, this coupled structural-alteration approach aims to develop a unifying deformation model for the investigated belt, identify key structural controls on hydrothermal alteration (and by inference Fe and Cu ± Au mineralization), and, to establish a new space-time framework for Svecokarelian-cycle deformation and hydrothermal fluid
flow in this sector of northern Fennoscandia.



## 2 Regional geological setting

The northern part of the Fennoscandian Shield is underlain by a continental nucleus of Archean (2.9-2.6 Ga) granitic, tonalitic and amphibolitic gneisses (Gaal and Gorbatschev, 1987; Weihed et al., 2005). In northern Sweden, Archean rocks belong to the Norrbotten Craton, one of three continental nuclei that were dispersed and reassembled during a 'Wilson-style' rifting and

accretionary-collisional cycle in the Paleoproterozoic (e.g. Lahtinen et al., 2005). Continental rifting and dispersal between c. 2.5 and 2.1 Ga developed crustal-scale rift-parallel fault systems and basins, voluminous tholeiitic mafic magmatism, and associated sedimentation which form a major large igneous province extending NW-SE from northern Norway to Russia (Pharaoh and Pearce, 1984; Hanski, 2012; Melezhik and Hanski, 2012; Hanski et al., 2014; Bingen et al., 2015). In northern Norrbotten, Rhyacian rift basins and related mafic igneous and sedimentary rocks occur within several NNE- and NNW-

trending greenstone belts (e.g. Martinsson, 1997; Melezhik and Fallick, 2010; Lynch et al., 2018a).

Early Svecokarelian-cycle orogenic magmatism (c. 1.90-1.86 Ga) in northern Sweden generated two regional suites of co-magmatic volcanic-plutonic rocks that are broadly divisible based on petrological, geochemical and geographical considerations. In the east, calc-alkaline series intermediate to felsic volcanic-volcanoclastic rocks and co-magmatic dioritic to granodioritic intrusions predominate (i.e. Porphyrite group and Haparanda intrusive suite of Bergman et al., 2001,

respectively). In the west, mildly alkaline (shoshonitic), intermediate to felsic volcanic-volcanoclastic rocks and co-magmatic monzonitic intrusions occur (i.e. Porphyry group and Perthtite Monzonite intrusive suite of Bergman et al., 2001). Late Svecokarelian-cycle magmatism from 1.81-1.65 Ga, during a possible phase of eastward subduction, generated wide spread I- to A-type granitic plutonism and coeval S-type granites from northern Norway to southern Sweden as part of the Transcandinavian Igneous Belt (Andersson, 1991; Åhäll and Larson, 2000; Weihed et al., 2002; Högdahl et al., 2004; Rutanen

and Andersson, 2009).

In general, metamorphic facies and related pressure-temperature (P-T) estimates are poorly constrained throughout northern Norrbotten (e.g. Skelton et al., 2018). However, based on metamorphic mineral assemblages and limited P-T modelling, Bergman et al. (2001) suggested the regional metamorphic grade increases from greenshist to amphibolite facies conditions going from west to east. East of the Western Supracrustal Belt (Fjällåsen; Fig. 1, 2, 3), syn-orogenic metavolcanic

rocks have yielded P-T values of 4.0 – 7.5 kbars and 630° - 805°C, respectively (i.e. amphibolite to granulite facies; Bergman et al., 2001). In the Gällivare area (Fig. 1), shear zone-hosted (mylonitic) schists along the Nautanen Deformation Zone have yielded P-T values of 2.5 – 4.3 kbars and c. 589 – 681°C, respectively (i.e. amphibolite to granulite facies; Tollefsen, 2014). Also in the Gällivare area, Romer (1996) reported U-Pb ages at 1730 Ma for fracture-hosted stilbite in metavolcanic rocks, suggesting the area remained below the closure temperature of stilbite (150° C) since then.

Paleoproterozoic rocks in northern Norrbotten record evidence of a complex, polyphase deformation history that evolved predominantly in response to Svecokarelian-related orogenic events (e.g. Vollmer et al., 1984; Forsell, 1987; Wright, 1988; Bergman et al., 2001; Bauer et al., 2018; Grigull et al., 2018; Luth et al., 2018). In the Kiruna area, Wright (1988) argued for an early $D_1$ thrusting event overprinted by gentle, local folding and shear zone development. Bergman et al. (2001) reported



two regional deformational events in northern Norrbotten and Bauer et al. (2018) showed a higher-grade deformation event overprinted by lower-grade folding in the Gällivare area. The timing of the early tectonothermal event has been suggested to 1.90-1.88 Ga by Cliff et al. (1990) based on a zircon U-Pb TIMS age for an undeformed granophyre dyke cutting the Kiirunavaara IOA deposit interpreted to represent the maximum age of the deformation. A similar timing of the same event

has been inferred based on deformation styles recorded by ca 1.89 – 1.88 Ga plutonic rocks in northern Norrbotten (Bergman et al., 2001). The timing of the later tectonic event is generally constrained by the emplacement of ca 1.8 Ga granitic intrusions and related hydrothermal activity (e.g. Bergman et al., 2001; Smith et al., 2009; Bauer et al., 2018).

## 3 Geology of the Western Supracrustal Belt

### 3.1 Setting, extent, and lithotypes

The Western Supracrustal Belt (WSB) refers to a discontinuous, c. 6 km-wide by 90 km-long, NNW-trending Orosirian (c. 1.89 – 1.87 Ga) lithotectonic domain located to the west of Kiruna in northwestern Norrbotten (Fig. 2, 3). In an earlier study, Wright (1988) defined the WSB as a north-south trending supracrustal inlier zone immediately to the west of Kiruna (i.e. the Eustiljåkk key area: Fig. 2, 3). However, this area represents the northernmost part of a much larger supracrustal terrain that extends further southward to the west and southwest of the Kiruna and Gällivare mining areas. In this study, we retain the

original nomenclature of Wright (1988) but expand the term "Western Supracrustal Belt" to include the areas from Allavaara-Fjällåsen in the south to Eustiljåkk in the north underlain by Orosirian metasupracrustal rocks (Fig. 2, 3). Similar lithostratigraphic domains occur to the west of the WSB as relatively small inlier "windows" surrounded by Paleoproterozoic plutonic rocks or younger (Paleozoic) Caledonian-cycle rocks (e.g. Angvik, 2014).

In general, the geology of the WSB is dominated by calc-alkaline to alkaline, volcanic-volcaniclastic rocks with basaltic

to rhyolitic compositions that were metamorphosed to approximately epidote-amphibolite facies conditions (Ros, 1979; Bergman et al., 2001; Edfelt et al., 2006). Along its margins, the WSB is bounded by subordinate c. 1.88-1.86 Ga granodiroritic to dioritic plutonic rocks and more abundant c. 1.80 Ga granitoids (Bergman et al., 2001). The plutons intrude, truncate and disrupt the supracrustal pile and this aspect, combined with the polydeformed nature of the sequence (see below), makes lateral stratigraphic correlations difficult. In the Ekströmsberg area (Fig. 2, 3), Rhyacian greenstones are found at the margin of the

belt providing a partly persevered pre- to syn-orogenic stratigraphic record (Offerberg, 1967; Witschard, 1975a). In the Allavaara area (Fig. 2, 3), Witchard (1975a) also indicated synclinal folds comprised of Rhyacian greenstones on their flanks cored by Orosirian felsic to intermediate volcanic rocks.

### 3.2 Structural geology

Previous studies incorporating parts of the WSB have provided an intermittent and somewhat contradictory assessment of its

structural character and evolution (c.f. Wright, 1988; Bergman et al., 2001; Edfelt et al., 2006). In the Eustiljåkk area (named Ruojtatjåkka South in Wright, 1988) in the north, Wright (1988) identified a steep, NW-trending mylonite zone that mimics





the NNW-orientations of high-strain zones at Allavaara in the south (Fig. 2, 3). The Eustiljåkk mylonite provided kinematic evidence for a west-side-down oblique normal sense-of-shear, based on rotated porphyroclasts with asymmetric tails (Wright, 1988). In contrast, Bergman et al. (2001) reported overall west-side-up kinematics for the composite shear zone within the WSB based on outcrop observations west of Kiruna and Gällivare (Fig. 1). A set of ENE-trending dextral strike-slip shear

zones in the Eustiljåkk area (Ruojtatjåkka South in Wright, 1988) have also been reported by Wright (1988) who suggested these structures post-date the dominant NNW-SSE tectonic grain.

Based on airborne (Bergman et al., 2001) and ground magnetic data (Frietsch et al., 1974), several prominent NNW-trending linear magnetic anomalies occur along the WSB, or as splay anomalies extending NW to WNW toward the Tjårrojåkka area (Fig. 3). These magnetic lineaments have generally been assigned to an unnamed, crustal-scale

Paleoproterozoic shear zone analogous to the major NE-trending Karesuando-Arjplog Deformation Zone to the northeast and the NNW-trending Nautanen Deformation Zone to the east (e.g. Bergman et al., 2001; Sandrin and Elming, 2006). Moreover, the magnetically anomalous character of the WSB mimics similar "striped" magnetic signatures associated with intense magnetite alteration and mylonitic deformation within the IOCG-mineralized Nautanen Deformation Zone near Gällivare to the east (Fig. 1; e.g. Lynch et al., 2015; Lynch et al., 2018b).

### 3.3 Mineralization and related alteration

Both iron oxide-apatite (IOA) and Cu ± Au mineralization occur along the WSB. The best documented examples are the Tjårrojåkka Fe-Cu system in the southwest and the Ekströmsberg IOA deposit in the north (Fig. 2, 3). The Tjårrojåkka system (Edfelt et al., 2006) comprises a western IOA deposit and an IOCG-style Cu ± Au body in the east. The IOA deposit is primarily associated with pervasive albite + scapolite + magnetite ± amphibole alteration, while "red rock"-style potassic-ferroan (K-

feldspar + hematite ± albite) alteration is mainly associated with the Cu deposit (Edfelt et al., 2006). The Ekströmsberg deposit comprises several parallel NW-trending magnetite and hematite ore bodies. The ore bodies are associated to sericite + quartz altered host rocks and discordant calcite veining, as well as muscovite, zircon, epidote, tourmaline and allanite (Frietsch, 1974).

In general, the siting of Fe and Cu mineralization along the WSB appears to be partly controlled by superimposed structures formed during polyphase deformation. In the Tjårrojåkka area, Edfelt et al. (2006) reported three deformational events; $D_1$ and

$D_2$ which generated cleavages in NE- and E-oriented shear zones, respectively, and a later $D_3$ event which folded $D_1$ structures and produced shallow SE-striking cleavages dipping towards the southwest. Additionally, the Fe and Cu ore bodies at Tjårrojåkka are aligned with $D_1$-related NE- to ENE-oriented planar structures (Edfelt et al., 2006). At the Ekströmsberg IOA deposit, Frietsch (1974) reported several prominently developed structures including NW-trending schistosity that parallel the orientation of the main ore bodies. Additional structural components including locally developed folding, a major NW-SE-

trending fault zone and tectonically crushed feldspar phenocrysts were also noted (Frietsch, 1974). Overall, these features imply a polyphase structural evolution for the Ekströmsberg IOA deposit but further detailed descriptions of its structural characteristics are presently lacking.



## 4 Methodology and study approach

In this study, five key areas were chosen to elucidate the structural differences and/or similarities along the WSB; from Eustiljåkk, Ekströmsberg, Tjårrojåkka, Kaitum West and Fjällåsen-Allavaara (Fig. 2, 3). Geological mapping with a structural focus was conducted between 2015 and 2017. A total of 698 outcrop observations were made and 1079 structural
measurements were collected. The mapping campaign covered all major outcrop areas between Allavaara in the south to Eustiljåkk in the north (Fig. 1, 2). All structural measurements were collected using Brunton Geo Pocket Transits and all data were digitized in the field on ruggedized iPad mini devices using the Middland Valley application Field Move (Midland Valley Exploration Ltd.). All lineations were measured as the pitch on planes and recalculated into true orientation using the software Geo Calculator (Holcombe, Coughlin, Oliver, Valenta Global). For magnetite-rich rocks, structural measurements were
estimated using known distal points in the terrain. Structural analysis was performed using the Move 2017 software package (Midland Valley Exploration Ltd.) and Leapfrog Geo 4.0 (Seequent), whereas maps were constructed using ArcMap (ESRI). Stereographic plots were produced as lower-hemisphere, equal-area stereographic projections using Dips 7.0 (Rocscience). Forty-one oriented samples were collected throughout the area. The samples were cut across foliation and parallel to lineation and sent to Vancouver Petrographics Ltd. for thin section preparation, one thin section per sample. Petrography and
microstructural investigations were performed using a conventional petrographic polarization microscope equipped with a digital camera (Nikon ECLIPSE E600 POL).

The characterization of hydrothermal alteration was approached from a "field geology" or exploration perspective. We focused on the recognition of mappable alteration mineral assemblages at the outcrop- to hand specimen-scale to establish possible links between certain structures and alteration styles, and identify specific structural-alteration combinations that may
prove useful as a vectoring tool toward Fe and/or Cu mineralization. The purpose of this approach was to provide a holistic overview of paired deformation-hydrothermal processes affecting the WSB, and offers a starting point for further alteration-related studies in this underexplored area.

## 5 Results

In general, the structural mapping results highlight several superimposed ductile and brittle-ductile deformation events along
the WSB that vary in terms of their character and intensity. Likewise, variably developed hydrothermal alteration displays localized differences in terms of type, style and intensity. In Sections 5.1 – 5.5, the main structural features from the five key areas are presented from north to south, with a description of the alteration characteristics for the WSB outlined in Section 5.6.

### 5.1 Eustiljåkk area

The Eustiljåkk area provides a relatively continuously exposed profile cross the WSB (Fig. 4). The area predominantly
comprises weakly deformed porphyritic volcanic rocks, along with subordinate metasedimentary rocks and mafic dykes. West-dipping shear zone-type structures occur in the NE-part of the area and impart a dominant N-S-directed structural grain in this



sector (Fig. 4). Beyond this area to the west, other large-scale ca NW-aligned structures are interpreted from magnetic anomaly data (Bergman et al. 2001, Fig. 3). Although ground truthing of these more western structures was not possible due to poor exposure, their continuity was verified by structural measurements and thin section analysis of similarly deformed rocks in key areas south of Eustiljåkk (Sections 5.2, 5.4 and 5.5 below).

In general, a weakly developed penetrative cleavage is present throughout the Eustiljåkk area. We designate this cleavage as $S_1$ because it is folded into mesoscale near vertical-plunging $F_2$ folds (Fig. 4: further descriptions below). NNW-SSE or N-S-trending and west-dipping grains that are parallel with magnetic lineaments appear to control the orientation of $S_1$. The magnetic lineaments predominately have NNW-SSE orientations (Fig. 3) whereas the stereographic projection of $S_1$-foliations from the entire Eustiljåkk area indicates a dominantly N-S grain (Fig. 5). This probably results from a bias in the $S_1$ summary

plot (Fig. 5a-b) due to a greater surface exposure of N-S directed magnetic lineaments in the area which lead to a higher number of structural measurements along this grain.

     $S_1$      is      defined      by      a      preferred      mineral      orientation      of feldspar + quartz ± biotite ± actinolite ± hornblende in felsic to intermediate volcanic rocks (Fig. 6a) and adjacent granitoids. The cleavage distribution in the granitoids is unevenly distributed and is generally of low intensity. South of the Eustiljåkk

area, calcite is also part of the cleavage mineral assemblage within porphyritic volcanic rocks. In mafic rocks, $S_1$ is defined by actinolite + plagioclase ± epidote ± hornblende ± calcite that show a preferred mineral orientation parallel to the ca N-S-aligned $S_1$ fabric.

     In general, bedding surfaces ($S_0$) are only rarely preserved in the Eustiljåkk area. In the southern part (Fig 4), deformed meta-arkosic horizons are interbedded with more competent volcanic porphyries. The more western of these horizons is

slightly steeper and shows intense NNE-verging parasitic $F_1$ folding with M- and Z-geometries (Fig. 6b). Based on bedding-bedding and bedding-cleavage relationships combined with parasitic fold geometries, we interpret the fold as a NE-verging, overturned synform. An axial planar-parallel $S_1$ cleavage generally dips sub-parallel to but is slightly steeper than bedding (Fig. 4). The calculated $F_1$ beta axis (β: 330/15) plunges gently towards the northwest, which is comparable to the measured fold axes of parasitic $F_1$ Z-folds (Fig. 4). The fold limbs are transected by an E-W-aligned high-strain zone inferred from the

aeromagnetic map and supported by high-strain cleavage measurements near the magnetic lineament (Fig. 4). The axial planar $S_1$ cleavage appears to be transposed into the high-strain zone (Fig. 4). We interpret this phenomenon as localized folding (transposition) of $S_1$ and $S_0$ in response to $D_2$ shearing. In the northwestern part of the Eustiljåkk map sheet, a low-intensity $S_1$ cleavage is folded into $F_2$ folds. Stereographic analysis indicates slightly asymmetrical fold geometries with a calculated beta axis plunging steeply (β:293/79) towards the northwest (Fig. 4). In terms of overprinting structures, $D_2$-related fold hinges ($F_2$)

and axial planar-parallel cleavages ($S_2$) were not observed in the field nor in thin sections.

     The northeastern part of the Eustiljåkk area (Fig. 4) is characterized by mainly N-S and subordinate NW-SE and NE-SW-trending sets of thin, well-exposed high-strain zones. Associated stretching lineations show variable orientations (Fig. 4). In low strain units, $S_1$ cleavages plot near the best fit plane (Fig. 4) suggesting $F_2$ folding around a calculated beta axis plunging steeply to the north (β:357/79), whereas the high-strain $D_2$ fabrics are oriented consistently N-S (Fig. 4). Mafic dykes are



commonly encountered either within or at the contacts of the high-strain zones (Fig. 6c). The mafic dykes sometimes show internal folding of leucocratic material but relatively undeformed dykes are also present. The mafic dykes are in most cases not wider than a couple of meters and occur as swarms. Where dyke density is high, the dykes were mapped as single units. The high-strain zones and related dykes affect a surrounding granitoid, implying granitoid emplacement occurred prior to $D_2$-

related deformation.

Oriented thin section samples from the Eustiljåkk area did not yield any high confidence kinematic indicators making the kinematic interpretation of the area difficult. Despite this, an east-verging $F_1$ fold in the southern part of the area (Fig. 4) indicates that the Eustiljåkk area was affected by west-side-up movements during $D_1$.

## 5.2 Ekströmsberg area

The Ekströmsberg area (Fig. 7) is dominated by felsic to intermediate metavolcanic rocks that host the Ekströmsberg IOA deposit (Frietsch, 1974). The dominating NNW-SSE trending high-strain zones west of the Ekströmsberg IOA deposit (Fig. 8) is poorly exposed and is mainly inferred from aeromagnetic and ground magnetic anomalies (Frietsch et al. 1974, Bergman et al., 2001). These magnetic anomalies can be linked to high-strain zones mapped just northwest of the Ekströmsberg IOA deposit (Fig. 8) and further south in the Kaitum West area (Fig. 11, Section 5.4).

At Ekströmsberg, a penetrative, continuous cleavage is defined by a preferred mineral orientation of feldspar + quartz and/or biotite or amphibole (mean orientation: 251/50). The orientation of this cleavage appears to be controlled by adjacent high-strain zones, hence suggesting this cleavage as $S_1$ generation. This fabric is developed in both the supracrustal and plutonic rocks in the Ekströmsberg area, although, ductile deformation only sporadically occurs in the plutonic rocks. A stretching lineation is generally present in felsic-intermediate volcanoclastic rocks and is defined by stretched feldspar

porphyroclasts and quartz. Occasionally, a very well developed mineral lineation ($L_m$) in mafic rocks is defined by strained actinolite-tremolite forming distinct L-tectonites (Fig. 8a). Nevertheless, the general trend in the area is the development of LS-tectonites.

Metasedimentary rocks are rare at Ekströmsberg and few clear bedding planes could be observed. Sedimentary rocks are only present in the northwestern part of the area, comprising poorly sorted, polymict conglomerate horizons with poorly

developed bedding (Fig. 7). Locally, pillow basalts provide reliable bedding markers and where magmatic layering is indicated by stratiform scapolite replacement (Fig. 8b). Due to the general lack of high confidence bedding markers few well exposed fold shapes has been observed. Lithological contacts (Offerberg, 1967) and magnetic maps (Frietsch et al., 1974, Bergman et al., 2001) were used to indicate fold symmetries where lithological contacts and magnetic lineaments bend in association with small-scale fold axes could be observed and measured in field (Fig. 7). $S_1$ cleavages are typically developed axial planar-

parallel to these inferred $F_1$ folds. A low intensity foliation of a somewhat uncertain origin is commonly observed in porphyritic volcanic rocks (Fig. 8c, d). This structure is defined by a preferred orientation of feldspar and quartz which resembles a tectonic cleavage. Nevertheless, no or very little signs of pressure shadows or re-crystallization around the feldspars could be observed (Fig 11). We suggest this fabric as a flow fabric implying that a primary magmatic foliation ($S_0$) may be discernible in the



Ekströmsberg area and that this foliation correlates to what Frietsch (1974) described as "fluidal banded" (Cc. Fig. 8 in Frietsch, 1974). This interpretation is supported by steeply plunging ($\beta$: 195/75), near cylindrical, tight and upright folding of this $S_0$ fabric whereas the clearly tectonic, continuous, low intensity $S_1$-cleavage does not show this folding pattern (Fig. 8).

The $S_1$ cleavage appears to be transposed into a set of sub-vertical, E-W-trending strike-slip- and NNW-SSE-trending dip-
slip shear zones near the Ekströmsberg IOA deposit. We interpret this as transposition of $S_1$ cleavage into $D_2$ shear zones implying that, as in the Eustiljåkk area, two compressional events can be recognized (i.e. $D_1$ and $D_2$). A mafic volcanic rock in the northwestern part of the area shows a crenulation cleavage developed in a NNW-SSE-trending dip-slip mylonitic zone, also suggesting two fabric-forming events (Fig. 8e).

The sense of shear determined from SCC´-fabrics in thin section indicates oblique west- to southwest-side-up kinematics
in the dominating NNW-SSE-trending high-strain zones. Oblique sinistral (8f) and reverse dip-slip (8g) movements are recorded along north- and near-vertical-plunging stretching lineations respectively, both yielding overall west-block-up kinematics. The E-W-trending mylonites within the Ekströmsberg IOA deposit (Fig. 7) show strike-slip movements with a sinistral top-to-the-west sense-of-shear as indicated by S-C-fabric orientations (Fig. 8h). Interestingly, many of the feldspar porphyroclasts in the E-W high-strain zones at the Ekströmsberg IOA deposit record brittle deformation (Fig. 8i). No brittle
deformation of feldspars was observed in the NNW-SSE-trending mylonites throughout the entire WSB. No direct cross-cutting relationships between the E-W- and NNW-SSE-trending high-strain zones could be observed in the field. However, an E-W tectonic grain offsets the NNW-SSE grain on magnetic maps (Bergman et al., 2001; Frietsch et al., 1974) throughout the WSB, suggesting the late timing of these high-strain zones on a regional to belt scale. Both structural grains show similar sub-grain rotation (SGR) and quartz recrystallization textures with local bulging (BLG) recrystallization (Cf. Fig. 8e, f, g, h)
indicating relatively low-temperature conditions (c.f. Passchier & Trouw 2005).

### 5.3 Tjårrojåkka area

The Tjårrojåkka area (Fig. 9) is dominated by mainly felsic and intermediate volcaniclastic, volcanic and volcanosedimentary rocks that host the Tjårrojåkka Fe-Cu system (c.f. Edfelt et al., 2005). The area is dominated by a penetrative $S_1$-foliation defined by the mineral alignment of amphibole, quartz, feldspar and locally magnetite. Bedding ($S_0$) is locally visible in
laminated volcanosedimentary rocks and greywackes. $S_0$ is generally sub-parallel to $S_1$ and shows local, meso-scale, intrafolial isoclinal folds (Fig. 10a). At the macro-scale, $S_0$ appears to form an isoclinal $F_1$-fold with an approx. ENE-WSW-trending main structural grain (Fig. 9). Mineral lineations are observable as an alignment of amphibole and quartz on $S_1$-foliation planes. Both bedding and $S_1$ foliations are overprinted by $F_2$-folds with axial traces trending approx. N-S to NE-SW (Fig. 9). $F_2$-fold geometries are upright, open to close and show a distinct, axial surface parallel spaced cleavage, here designated $S_2$. The N- to
NE-aligned $S_2$-cleavage is commonly defined by biotite alignment and also by brittle fracturing. Spacing of the $S_2$-cleavage ranges from a few cm up to several tens of cm (Fig. 10b, c). Locally, $S_2$-related kink bands are relatively common features in volcaniclastic rocks (Fig. 10 b). In general, the dominate NE-trend of $S_1$ mimics the orientation of a NE-SW-striking high-strain zone hosting the Tjårrojokka copper-gold deposit (c.f. Edfelt et al., 2005, 2006).



### 5.4 Kaitum West area

The Western Supracrustal Belt progressively widens southward toward the Kaitum West area (Fig. 2, 11). Overall, the bedrock in this area is dominated by felsic to intermediate metavolcanic to volcanoclastic rocks with additional relatively common metabasalt sequences. Kaitum West provides an almost continuous mappable E-W profile across the WSB. Structural observations indicate the area constitutes several low-strain zones that border a central high-strain block (Fig. 11). The main strain was taken up by shear zones concentrated in felsic volcanoclastic rocks. The surrounding metabasaltic to meta-andesitic volcanic rocks show lower strain intensities and the easternmost low-strain block shows $F_2$ folding around a calculated beta axes plunging steeply to moderately towards south (β: 182/67: Fig. 11). We interpret the central high-strain block as the northern continuation of the shear zone system mapped in the Fjällåsen-Allavaara area (see section 5.5). Within the high-strain block, several NW-SE-, N-S-, and locally E-W-trending high strain zones and shear zones are present (Fig. 11). The shear zones are best developed in volcanoclastic rocks, although highly strained intermediate and mafic volcanic rocks also occur.

Bedding markers are rare in the Kaitum West area, however, local volcanosedimentary horizons show bedding markers. Within the high-strain block, a polymict and poorly sorted clastic horizon occurs (Fig. 12a, b). This horizon shows an $S_1$ cleavage (266/79) developed as sericite, biotite, chlorite dominated shear bands sub-parallel to bedding and is particularly well developed at the margins of compositional layers (Fig. 12a). At outcrop-scale, the $S_0/S_1$ composite fabric is isoclinally folded (Fig. 12b) around a measured fold axis (335/60) plunging moderately towards NW. Locally, a $S_1$ high-strain cleavage is transposed into shear bands that parallel the main shear zone directions (Fig. 12 c, d). This indicates relatively high-strain was localized both during $D_1$ and $D_2$ in the Kaitum West area.

Kinematic indicators in the Kaitum West area suggest southwest-side-up and shallow to steep oblique dextral displacements that are associated with southward-plunging stretching lineations. This interpretation is based on SCC´ fabrics and asymmetric quartz sigmoids with stair stepped pressure shadows (Fig. 12e). The same sense-of-shear is indicated by sinistral SC fabrics (Fig. 12f) along north plunging stretching lineations north of the Kaitum West area (south of Ekströmsberg; Fig. 2, 3) where the same shear zone system is related to a major fold structure. Locally, contradicting sense-of-shear (east-block-up) is indicated by asymmetric sigma clasts with poorly developed pressure shadows.

### 5.5 Fjällåsen-Allavaara area

The area between Fjällåsen and Allavaara is dominated by felsic, intermediate and mafic volcanic, volcaniclastic and volcanosedimentary rocks. The overall dominant structural grain is an approx. N-S-trending set of high-strain zones (Fig. 13) associated with a well-developed penetrative foliation. The foliation is defined by the alignment of strained amphibole, biotite, feldspar, quartz and locally magnetite (Fig. 14a) and is here assigned $S_1$. Bedding is locally observable as compositional layering in volcanosedimentary rocks and is typically sub-parallel to the $S_1$-foliation. Locally, isoclinally $F_1$ folded quartz and amphibole veins and bedding can be observed (Fig. 14b, c). Shearing is commonly observed and localized in prominent high-strain zones transposing $S_0$ and $S_1$ and forming distinct mylonites (Fig. 14d). Based on asymmetric sigma clasts and SC-fabrics





a reverse, west-block up sense-of-shear is interpreted for the majority of zones. The central shear zone in Fjällåsen hosts the Fjällåsen Cu-prospect and shows oblique kinematics with a reverse west-block up and sinistral sense-of-shear along a mineral lineation plunging 60° N (Fig. 13). $S_0$ and $S_1$ are folded openly to tightly into meso-and macro-scale $F_2$ folds (Fig. 14e, f). An axial surface parallel cleavage ($S_2$) is locally observable as a weakly developed, semi-brittle, spaced cleavage (Fig. 14e, f).

Locally, the $S_2$ cleavage shows en-echelon fracturing indicating a certain amount of brittle movement (Fig. 14b). Mineral lineations are variably plunging steeply to moderately towards the north and south.

## 5.6 Metamorphism and hydrothermal alteration along the WSB

In this section, descriptions of alteration mineral assemblages follow the qualitative approach of Gifkins et al. (2005) and are summarised in Figure 16. Alteration assemblages that were temporally resolvable based on cross-cutting or overprinting

relationships are classed as a separate assemblage. We recognise that these assemblages likely form part of a progressively evolving metasomatic-hydrothermal system but this approach was favoured to provide a field-based classification scheme for the various alteration occurrences.

In general, scapolite ± albite alteration is relatively common throughout the WSB and is mainly observable in compositionally mafic rocks (i.e. metabasalt, metadolerite) as a distinctive speckled (porphyroblastic), pale grey to creamy

white texture on exposed surfaces (Fig. 15a). Disseminated scapolite porhyroblasts are typically medium- to coarse-grained (1 – 8 mm), irregular tabular to elongate prismatic, and occupy ca 10 – 35 vol. % of altered units (Fig. 15a, c). The weakly to strongly developed porphyroblastic scapolite ± albite alteration is regionally distributed and best preserved in relatively low strain areas adjacent to or within NNW- to N-trending $D_1$-related shear zones. In the Eustiljåkk area (Fig. 2, 3), porphyroblastic scapolite alteration affects mafic dykes in zones with relatively high strain ($D_1$ and $D_2$), while it locally overprints inferred

bedding planes in basalt units in the Ekströmsberg area (Fig. 2, 3, 8b). Discordant, vein-related scapolite ± albite alteration also locally occurs and is paragenetically late relative to patchy porphyroblastic scapolite ± albite, although these veins are affected by shearing with a probable $D_2$ timing (Fig. 15b). Scapolite ± albite alteration overprints pervasive magnetite + amphibole alteration but is also locally overprinted by irregular and patchy amphibole ± magnetite zones that tend to be developed along $S_1$ foliations (Fig. 15c). Thus, both assemblages are interpreted to have formed synchronously during $D_1$.

Discordant

magnetite + amphibole veins with white to buff albite haloes also occur (Fig. 15d) and appear to have formed synchronously with the more pervasive (disseminated) magnetite + amphibole alteration.

Relatively intense and pervasive tremolite-actinolite alteration occurs at one locality in the northern part of the Ekströmsberg area where it forms an amphibolite L-tectonite within a steep, reverse, dip-slip shear zone (Fig. 8a). This

assemblage is overprinted by syn-tectonic ($D_2$), relatively intense and pervasive calcite alteration and post-tectonic, undeformed, granular epidote (Fig. 15e). Syntectonic calcite alteration is also present several km north of this locality along the same structure. Intense and pervasive calcite alteration (Fig. 15f) also occurs in the nearby Vieto area (Fig. 2, 3) where it is related to a moderately (c. 285/75) west-dipping shear zone. Here, calcite alteration overprints ductile shear zone fabrics and





the timing of calcite is therefore interpreted as late- to post-$D_2$ deformation at this locality. Deformed and discordant calcite veins, with a probable $D_1$ timing, are also common throughout the WSB and are mainly present in low-strain blocks dominated by mafic rocks.

Epidote forms part of several alteration assemblages with differing parageneses and structural associations along the WSB. For example, epidote together with hornblende and plagioclase occurs parallel to $S_1$ fabrics (likely syn-D1) in Rhyacian metabasalts to form a probably regionally distributed, moderately intense and pervasive metamorphic key assemblage (Fig. 15g). Additionally, early developed (pre- to syn-$D_1$) hornblende + epidote veins locally occur and are affected by $S_1$ foliations (Fig. 15g). In the Eustiljåkk area, epidote is also commonly developed as a retrograde product replacing disseminated hornblende in mafic metavolcanic rocks. A relatively intense, selectively pervasive epidote + K-feldspar assemblage, spatially related to localized Cu-sulphide weathering overprints a weak, pervasive amphibole + magnetite alteration in the western low-strain block of the Kaitum West area (Fig. 15h).

The most prominent alteration assemblage in the Tjårrojåkka area is a relatively intense and pervasive K-feldspar + epidote + quartz hydrothermal alteration assemblage that overprints and cuts $S_1$ foliations. Locally, both K-feldspar and epidote appear to be remobilized into $S_2$ spaced cleavage domains. Epidote also commonly occurs as patches on fracture planes at Tjårrojåkka, producing a distinctive reddish-green rock (Fig. 15i). Epidote also occurs in areas of often moderate to intense, selectively pervasive K-feldspar alteration (replacing albite) throughout the WSB (Fig. 15j) and is typically accompanied by weak retrograde sericite alteration of secondary K-feldspar. Overall, this assemblage commonly affects intermediate to felsic metavolcanic rocks that may also display the effects of earlier formed scapolite and/or amphibole ± magnetite alteration where K-feldspar ± epidote is less intensely developed. These overprinting relationships and the more localized (selectively pervasive) distribution of the K-feldspar + epidote assemblage suggests it formed during a later alteration event (i.e. post-$D_1$).

In the Ekströmsberg area, a relatively intense, shear band-hosted biotite + magnetite + K-feldspar + muscovite assemblage affecting a rhyodacitic volcanosedimentary rock is associated with steep, approximately E-W-striking sinistral strike-slip shear zones (Fig. 15k, l). The biotite shear bands are oriented subparallel with volcanosedimentary bedding ($S_0$). Magnetite also occurs along the same shear bands and as disseminated grains in the adjacent wall rock. We interpret the timing of this alteration to be synchronous with the last activity of the structure, hence $D_2$ (see section 5.2).

## 6 Discussion

### 6.1 Structural evolution of the WSB

In general, the structural elements preserved within the WSB are consistent with a complex, polyphase deformation history. Observed $S_1$ cleavages are axial planar to folded $S_0$ bedding planes and magmatic flow structures associated with metasedimentary and volcanosedimentary rocks. $F_1$ folds are generally poorly exposed and are thus difficult to constrain due to the lack of clear bedding and/or stratigraphic younging indicators. Where developed, however, $F_1$ folds are tight to isoclinal and are either upright or overturned, with the latter verging to the east (Fig. 4, 7). Interpreted $F_1$ folding in the Ekströmsberg





area has yielded a relatively steep calculated beta axis towards the SSW (β: 195/75), which is a typical feature for the $F_2$-folds throughout the WSB. These similar orientations suggest earlier formed $F_1$ folds were rotated into a steep southward plunge during $D_2$ transposition.

$S_1$ cleavages throughout the WSB are best preserved in relatively low-strain domains as a penetrative fabric in supracrustal
rocks, while they are only weakly developed in adjoining shoshonitic plutonic bodies (equivalent to the Perthite monzonite suite (PMS) of Bergman et al., 2001). In contrast, calc-alkaline plutonic rocks of similar age (Haparanda suite) typically display well-developed planar cleavages in northern Norrbotten indicating these intrusions formed pre- to syn-$D_1$ (cf. Bergman et al., 2001). Based on the more intense foliations persevered in the calc-alkaline plutonic rocks compared to the shoshonitic intrusions, we interpret the relative timing of the latter as syn- to late-$D_1$. Reported radiometric igneous ages for various
intrusions assigned to this shoshonitic (PMS) plutonic suite in northern Norrbotten range from c. 1.88-1.86 Ga (Bergman et al., 2001, Sarlus et al., 2017, Kathol and Hellström, 2018).

Mylonitization frequently occurs subparallel with the regionally extensive and laterally continuous $S_1$ cleavages. Although the level of outcrop exposure does not allow for accurate estimations of the width of these zones, our mapping experiences combined with ground magnetic signatures in the Ekströmsberg area (Frietsch et al., 1974) indicate that these zones are
relatively thin (meters to tens of meters) and display sharp contacts controlled by lithological contrasts. The mylonitic strain seems to have been taken up by volcanoclastic and metasedimentary horizons sandwiched between more competent volcanic rocks throughout the WSB.

In general, $S_2$ cleavages developed predominantly in NNW-SSE- and approx. E-W-trending high-strain zones. Where $S_1$ is not completely transposed into parallelism or overprinted by high-strain fabrics, the $S_1$ cleavage is occasionally overprinted
by a $S_2$ crenulation cleavage (c.f. Fig. 8e). Dynamic quartz recrystallization textures formed during $D_2$ in the mylonite zones (SGR texture with minor BLG) suggest low to medium temperature conditions within these zones (Passchier and Trouw, 2005). In comparison, slightly higher temperature conditions are suggested for the Kaitum West area based on the presence of grain boundary migration (GBM) and SGR textures (Passchier and Trouw, 2005).

In the Tjårrojåkka area, Edfelt et al. (2006) identified three deformation events ($D_1$ – $D_3$ in Edfelt et al., 2006), although
detailed structural information for each of these was not given. In their study, Edfelt et al. (2006) reported an early brittle-ductile deformation event ($D_1$), which generated steep NE-SW-directed planar structures that controlled the siting and orientation of the Tjårrojåkka Fe-Cu system. Subsequently, E-W oriented shearing and folding ($D_2$ structures) that was temporally associated with localized NNW-SSE-trending deformation ($D_3$ in Edfelt et al., 2006). Based on our mapping results (Section 5 and Figs. 7, 9, 11), we interpret ca E-W-directed high-strain fabrics at Tjårrojåkka (equivalent to $D_2$ structures in
Edfelt et al., 2006) as relatively late structures (our $D_2$ event) since ca NNW-SSE-aligned fabrics are consistently affected (e.g. folded and displaced) by ca E-W-aligned structures throughout the WSB (Fig. 2, 3, 14a, 14d). In particular, the structural map of the Tjårrojåkka area shows isoclinal $F_1$ folds that are openly refolded by $F_2$ folds and offset by an E-W-directed $D_2$ high-strain zone (Fig. 9).





In low strain blocks throughout the WSB, $S_1$ planar structures are folded into meso-scale $F_2$ folds with fold axes generally plunging moderately to steeply (60-80°) northward or southward (Fig. 4, 9, 11). Axial planar-parallel $S_2$-cleavages related to the $F_2$-folds in the low strain blocks are very rare. Only a few examples in the Tjårrojåkka, Fjällåsen and north of the Ekströmsberg area have been observed where $S_2$ forms an axial planar-parallel, brittle-style, spaced cleavage (Fig. 10b, 10c,

14d). Therefore, we suggest $D_2$ deformation occurred during relatively low-pressure, upper-crustal conditions (c. f. Pfiffner, 2017). In the Gällivare area (Fig. 1), Bauer et al. (2018) interpreted folding of an $S_1$ gneissic fabric into $F_2$ synformal structures without axial planar $S_2$ cleavages as low-pressure, shallow crustal level deformation. In the Aitik Cu-Au-Ag deposit, Wanhainen et al. (2012) report lower amphibolite facies metamorphism and deformation at 500-600° and 4-5 kbar at 1.88 Ga. This medium-grade tectonothermal event was later overprinted by a hydrothermal event estimated to 200-500°C and 1-2 kbar

at 1.78 Ga (Wanhainen et al., 2012). The findings in the Aitik Cu-Au-Ag and Malmberget IOA deposits may not be directly applicable to the WSB in terms of metamorphic conditions but are comparable with respect to a pronounced earlier deformation event (regional $D_1$) overprinted by a weaker deformational event (regional $D_2$).

While the identification of two generations of planar fabrics is relatively straightforward in the WSB based on their orientations and interrelationships, linear structures are more difficult to interpret due to the lack of crosscutting relationships.

Stretching lineations measured on $S_1$ planes in low strain blocks are interpreted as $L_1$ structures. The orientation of $L_1$ lineations varies considerably more than stretching lineations measured in relatively high-strain zones (c. f. Fig. 17a-b). Crenulation of mylonitic cleavage has only been observed along near vertical stretching lineation, which lead us to interpret the well clustered near-vertical stretching lineation (Fig. 17b) as $L_2$. The sense-of-shear associated with sub-vertical $L_2$-lineations is reverse dip-slip (Fig. 8g) and best explained by an E-W compressional stress field (see also section 6.1.4). Shallow east-plunging lineations

(Fig. 17b) were measured on relatively steep, approx. E-W-oriented planes offsetting the NNW-SSE grain, suggesting these structures as $D_2$-related structures. Sinistral strike-slip movements along the shallow east-plunging lineation cluster (Fig. 17b) in the Ekströmsberg area suggests these movement to have occurred as a response to E-W-directed crustal shortening and the lineation cluster is interpreted as $L_2$ (see section below). We suggest that the non-clustered shallow to moderately north and south plunging stretching lineations (Fig. 17b) within the NNW-SSE-trending mylonites might represent traces of an early

formed $L_1$-lineation within these high-strain zones. The sense-of-shear along these inferred $L_1$-lineations is reverse oblique-slip SW-side up and best explained to result from NE-SW-directed crustal shortening. This implies that the kinematics of $D_1$ and $D_2$ are best explained by two compressional events that deviate approx. 45° from each other. Based on the assumption that traces of $L_1$ can be identified, we argue that the steep to near-vertical cluster in the low strain $L_1$-plot (Fig. 17a) might represent $L_1$ lineations that were subsequently transposed during $D_2$ in a similar manner as $F_1$-fold axes were transposed in the

Ekströmsberg area (discussed in Section 5.2 above).

The NNW-SSE-directed mylonites containing moderately plunging stretching lineations ($L_1$) suggest oblique-slip SW-side-up kinematics based on SCC´ fabrics and rotated porphyroclasts in oriented samples from the Ekströmsberg, Kaitum West, and Fjällåsen-Allavaara key areas. These kinematic indicators suggest both sinistral and dextral movements occurred, with dextral movements recorded along moderate S-plunging lineations (Fig. 12e) and sinistral movements along moderate N-



plunging lineations (Fig. 8f, 14c). Similar kinematics are indicated during $D_1$ by the east verging $F_1$-fold in the Eustiljåkk area implying consistent reverse oblique-slip southwest-side-up movements during $D_1$ throughout the WSB.

The kinematics derived from S-C fabrics along the near vertical lineations ($L_2$ generation) within the NNW-SSE-oriented mylonites in the Ekströmsberg and Fjällåsen areas indicate reverse dip-slip, W- to WSW-side up sense-of-shear (Fig. 8g). This
implies a reactivation of the NNW-SSE-trending structures during an approx. E-W-directed $D_2$ shortening. Additionally, sinistral strike-slip movements along E-W-trending shear zones in the Ekströmsberg area (Fig. 8h) indicate they were active during ca E-W compression and coincident with reverse dip-slip movements along the NNW-SSE-trending mylonites. A late timing of the E-W-trending structures is supported by the consistent offset of NNW-SSE-directed grain by E-W-directed structures throughout the WSB (Fig. 3).

**6.2 Summary of major deformation events affecting the WSB**

Based on the new structural data presented in this paper and with reference to tectonic models proposed for surrounding areas (Wright, 1988; Talbot and Koyi, 1995; Bergman et al., 2001; Angvik, 2014; Skyttä et al., 2012; Andersson et al., 2017; Sarlus et al., 2017; Grigull et al., 2018; Luth et al., 2018; Lynch et al. 2018) , we propose the following tectonic model that utilizes two major deformation events for the WSB (Fig. 18):

**6.2.1 $D_1$-event**

NE-SW to ESE-WNW-directed crustal shortening (this study; Wright, 1988; Talbot and Koyi, 1995; Lahtinen et al., 2005; Angvik, 2014), with syn-tectonic plutonism at 1.88-1.86 Ga (Bergman et al., 2001; Sarlus et al., 2017, Kathol, and Hellström 2018), generated ENE-verging, tight to isoclinal, shallowly NNW-plunging folds and a regionally distributed, consistently steep, WSW-dipping $S_1$-cleavage. This interpretation is mainly based on the $F_1$-fold patterns preserved in the Eustiljåkk area
(Fig. 4). These fold structures could have formed in a fold-and-thrust belt (Wright, 1988; Talbot and Koyi, 1995; Angvik, 2014) or, alternatively, during basin inversion (Andersson et al. 2017). Crustal thinning during the emplacement of the volcanic rocks of the WSB has been inferred by petrological/geochemical studies (Perdahl and Frietsch, 1993; Martinsson, 2004; Sarlus et al., 2017, 2018) and assigned to a backarc basin developed during early Orosirian times (Sarlus et al. 2017, 2018). Furthermore, the Orosirian stratigraphic record in central Kiruna area indicates the existence of a sub-basin formed in response
to this extensional setting (Andersson et al. 2017).

Oblique-reverse dextral and sinistral shear zones with west-side-up sense-of-shear were developed during $D_1$ and produced the dominant NNW-SSE-aligned, undulating magnetic lineaments (Bergman et al., 2001) that characterize the WSB. The sense of shearing is based on SCC´ fabrics along shallow to moderately plunging stretching lineations ($L_1$) along the NNW-SSE grain in the Ekströmsberg, Kaitum West and Fjällåsen-Allavaara areas. Activity along these zones during $D_1$ is indicated by
the presence of crenulated $S_1$ (Fig. 8e) and high-strain $S_1$-fabrics that are tightly folded in response to shearing along $D_2$ shear bands parallel to the NNE-SSW structural grain (Fig. 12c-d). The strain was taken up by rheological/lithological contrasts (or pre-existing discontinuities?), and favorable, incompetent lithologies such as volcanosedimentary rocks.

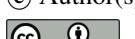



Based on the steep dip of the NNW-SSE grain, the syn-extensional geochemical character of the volcanic rocks (Perdahl and Frietsch, 1993; Bergman, et al., 2001; Martinsson, 2004; Martinsson et al., 2016; Sarlus et al., 2017, 2018), and the epidote-amphibolite facies (this study; Ros, 1979; Edfelt et al., 2005) recorded by volumetrically minor Rhyacian pillow lavas in the Ekströmsberg area, we favor a model involving inversion of a backarc basin to account for $D_1$-related structures in the WSB.

This contrasts with models that envisage the development of a classic fold-thrust belt in the WSB and Kiruna area during $D_1$ (Wright, 1988; Talbot and Koyi, 1995). Consistent evidence for the rotation of originally shallow-dipping thrust-type structures into sub-vertical orientations was not found along the WSB. Such a model was favored by Wright (1988) and rejected by Bergman et al. (2001). Furthermore, no classic fold-thrust belts involving shallow thrusts have been identified in nearby areas (Vollmer et al., 1984; Wright, 1988; Talbot and Koyi, 1995; Grigull et al., 2018; Luth et al., 2018). However, Angvik (2014)

identified a series of fold-thrust belts in the Rombak Teconic Window, west of the WSB. It is possible that the Rombak Tectonic Window represents a fundamentally different setting within the same volcanic arc environment and that the change from a fold-thrust to an extensional back-arc setting is to be found in-between the WSB and the Rombak Tectonic Window.

### 6.2.2 $D_2$-event

A phase of ca E-W compression caused meso-scale folding of $S_1$ foliations and produced near-cylindrical, upright and steeply

N- and S-plunging $F_2$ folds. Similar $F_2$-folding characteristics are developed in all key areas throughout the WSB. Based on the general lack of axial planar-parallel $S_2$-cleavages, we suggest that $D_2$-related folding took place at relatively shallow crustal levels and/or low-pressure shortening conditions (Pfiffner, 2017).

Strong strain partitioning focused $D_2$-related deformation into pre-existing NNW-SSE-aligned $D_1$-oblique-slip shear zones causing their reactivation with a reverse dip-slip, west-side-up sense-of-shear. Synchronously, near vertical E-W directed

sinistral (and dextral?: Wright, 1988) brittle-ductile strike-slip shear zones were active and off-set the NNW-SSE grain. Applying a basin inversion model to the WSB implies that the E-W directed structures might have originated as transfer faults between NNW-SSE trending normal faults and that the combined structural configuration was reactivated together, first during $D_1$ and later during $D_2$.

$D_2$-related kinematics are based on SC fabrics along steep- to near-vertical plunging $L_2$ stretching lineations within NNW-

SSE-trending high-strain zones at Ekströmsberg, north of Kaitum West and the Fjällåsen-Allavaara areas, and also  along shallowly E-plunging stretching lineations from E-W-trending high-strain zones in the Ekströmsberg area. The E-W-directed offset of NNW-SSE-trending high-strain zones is interpreted from magnetic maps (Frietsch et al., 1974; Bergman et al., 2001). Joints and fracture planes pre-date the latest epidote alteration in the area and are interpreted as developed either during $D_2$ or slightly after.

### 30  6.3 Timing of D1-D2 deformation within the WSB, and comparative links with adjacent areas

Tectonic models for northern Norrbotten and the Skellefte district generally include an early phase of deformation at approx. 1.88-1.86 Ga (e.g. Wright, 1988; Talbot and Koyi, 1995; Lahtinen et al., 2005; Skyttä et. al., 2012; Angvik, 2014). In the

 

Skellefte district, the timing of crustal shortening with related folding and shearing was constrained to 1.87 Ga (Skyttä et al., 2012), which is comparable to the 1895 Ma - 1877 Ma maximum age of NNW-SSE-trending shear zones and related 1886 Ma - 1837 Ma Au-Cu mineralization in the Kautokeino Greenstone belt (Bingen et al., 2015, Henderson et al., 2015). In accordance with this study, Allen et al. (1996), Bauer et al. (2011), and Skyttä et al. (2012) indicated that the shear zones formed during a

phase of arc-extension and volcanic activity prior to 1.88 Ga, and were subsequently reactivated during the 1.87 Ga accretion of the arc onto the Archean continent and related crustal shortening. Alternatively, in the West Troms Basement Complex some 250 km northwest of the WSB, Bergh et al. (2010) record no evidence of 1.88-1.86 Ga deformation, but instead a pronounced ductile footprint developed at 1.80-1.75 Ga in response to NE-SW shortening.

Overall, the interpreted early history of the NNW-SSE tectonic grain in this study is similar to structural mapping results

from the Skellefte district further south (Skyttä et al., 2012) where an approx. N-S-directed crustal shortening ($D_2$ in Skyttä et al., 2012) was constrained to 1.87 Ga by SIMS U-Pb zircon dating. Such a timing of $D_1$ ($D_2$ in Skyttä et al., 2012) would harmonize well with our interpretation of $D_1$ as synchronously with the plutonic rocks (1.88-1.86: Bergman et al. 2001) boarding the WSB.

In terms of later deformation events, Bauer et al. (2016) and Lynch et al. (2018) report west-side-up movements during a

$D_2$ phase of deformation in the Nautanen Deformation Zone (NDZ) near Gällivare which generated a duplex Riedel-shear system within that composite zone. Additionally, Bauer et al. (2018) argue for E-W compression during a $D_2$ phase of deformation and link this event to the intrusion of 1.8 Ga syn-tectonic minimum melt granites (e.g. Öhlander et al., 1987; Bergman et al., 2001; Sarlus et al., 2017). The results from the NDZ agree with results from the Rombak Tectonic Window (Angvik, 2014) ca. 100 km west of the WSB. Angvik (2014) brackets the timing of a comparable event ($D_3$-$D_4$ in Angvik,

2014) between 1778 Ma and 1798 Ma based on U-Pb zircon ages for syn-tectonic granites. Based on the above studies, we suggest a similar timing for $D_2$ in this study, which includes folding, reverse dip-slip reactivation of NNW-SSE-directed $D_1$ shear zones, and strike-slip shearing along E-W directed structures under brittle-ductile conditions.

## 6.4 Hydrothermal alteration and its relationship to deformational events

A preferentially aligned hornblende + epidote + plagioclase assemblage defines the $S_1$ continuous cleavage in Rhyacian

metabasalts in the Ekströmsberg area (Fig. 15e). A similar mineral assemblage was used by Ros (1979) and Edfelt et al. (2005) to define the metamorphic grade affecting mafic rocks in the Tjårrojåkka area (Fig. 2, 3, 9). According to Spear (1993), hornblende + epidote + plagioclase would indicate a transition from greenschist to amphibolite facies conditions. Similarly, we interpret the hornblende + epidote + plagioclase assemblage as a key metamorphic indicator assemblage (Ros, 1979; Edfelt et al., (2005) which accords with the generally accepted, but poorly constrained, low- to medium-grade low-P regional

metamorphism of northern Norrbotten (e.g. Frietsch et al., 1997; Bergman et al., 2001; Skelton et al., 2018). In Figure 15g the hornblende + epidote + plagioclase $S_1$-fabric forms axial planes to a folded hornblende + epidote vein-fill and possibly indicates a hydrothermal origin and pre-compressional timing for some hornblende + epidote. $F_1$-folded hornblende-veins with axial planar-parallel $S_1$ foliation have also been observed in the Fjällåsen-Allavaara area (Fig. 14a, d).



Several generations of albite + scapolite are present throughout the WSB. The porphyroblastic and the semi-conformable (selectively pervasive) types (Fig. 15a, b) are commonly encountered in low strain blocks close to shear zones or in mafic dykes within or at the margins of meter-wide shear zones. Scapolite porhyroblasts are often undeformed and tend to overprint early fabrics. Hence, we infer the timing of the regional porphyroblastic scapolite formation as syn- to late-$D_1$. This inference

5 is broadly consistent with a metasomatic (titanite) age of ca 1.9 Ga reported for a scapolite-altered mafic dyke that forms part of a Rhyacian greenstone sequence in the Nunasvaara area of central north-central Norrbotten (Smith et al., 2009). In the NDZ to the east of the study area, Lynch et al. (2015) report early sericite + scapolite + feldspar, which is probably representing the same regional alteration event. Local, late and discordant occurrences of scapolite, mainly in veins and shear bands, frequently occur proximal to the early scapolite. This discordant scapolite probably represents remobilization of the early albite + scapolite

10 during possible $D_2$-related deformation which would support recycling of chlorine in northern Norrbotten as suggested by Bernal et al. (2017). Stable Br/Cl ratios of scapolite altered rocks in northern Norrbotten plot in the magmatic field and suggest a mainly igneous fluid source (Martinsson et al., 2016). However, several studies have postulated that scapolite occurrences in lower Rhyacian greenstones might represent former evaporate beds (Frietsch et al., 1997; Martinsson, 1997; Bernal et al., 2017). Nevertheless, due to a spatial relationship with 1.88-1.86 Ga monzonite (shoshonitic) plutons and mafic volcanic rocks,

15 regional scapolite alteration in northern Norrbotten has been attributed to have played a role in the formation of IOA and Cu-Au deposits (e.g. Frietsch et al., 1997; Martinsson et al., 2016).

 Pervasive amphibole + magnetite alteration (Fig. 15b, f) is commonly distributed in the WSB and probably of regional significance. Locally, it is overprinted and affected by both $S_1$ foliations and paragenetically later alteration assemblages (Fig. 15f), hence, we attribute the timing of amphibole + magnetite alteration as early- to syn-$D_1$. Pervasive amphibole + magnetite

20 occasionally shows an early timing relative to the more selectively pervasive albite + scapolite alteration (Fig. 15b), although we have also observed the opposite relationship. In this context, the possibility that both assemblages represent an evolving calcic-sodic-ferroan alteration system similar to that suggested in other analogous IOA-IOCG-mineralized terranes may also be valid (e.g. Corriveau et al., 2016; Montreuil et al., 2016 a, b). Similar crosscutting relationships have been documented in the Cu-Au-mineralized NDZ near Gällivare (Lynch et al., 2015), although Lynch et al. (2015) included biotite as part of an

25 selectively pervasive amphibole + magnetite alteration assemblage. Amphibole + magnetite alteration also occurs as discordant veins with albite haloes (Fig. 15c). The relative timing of these veins is difficult to resolve due to the lack of an association with other discernable tectonic structures but we tentatively assign a $D_1$ timing and a paragenetic link with the pervasive amphibole + magnetite assemblage based on their similar mineralogy and often close spatial relationship.

 Regional selectively pervasive (patchy) K-feldspar alteration (replacing albite) is an important component in the felsic

30 volcanic rocks of the WSB (Fig. 15g, h). It was documented already during the early mapping campaigns in the area (Offerberg, 1967) and is often accompanied by retrograde sericite and epidote. Development of selectively pervasive K-feldspar has a late timing relative to $D_1$-related deformation, hence, it is assigned a $D_2$-timing in this study. Local vein-fill K-feldspar alteration is rather common and is sporadically overprinted by epidote, iron oxides and sulphide (Fig. 15f). In the Tjårrojåkka area, Edfelt et al. (2005) suggest K-feldspar alteration is paragenetically late relative to scapolite. Similarly, in the Gällivare area, K-



feldspar alteration shows a late $D_2$-timing and a close spatial relation to c. 1.8 Ga granites and pegmatites (Bauer et al., 2018), hence in agreement with our observations from the WSB.

A late $D_2$-timing is also interpreted for the biotite + magnetite + K-feldspar alteration hosted by sinistral E-W shear zones near the NNW-SSE-aligned Ekströmsberg IOA deposit (Fig. 15i, j). A late $D_2$ timing of this potassic-ferroan alteration is evident by off-setting relationships between the E-W structural grain and more dominant NNW-SSE-aligned $D_1$-related fabrics. Magnetite in this shear band-hosted mineral assemblage may be locally remobilized from the IOA deposit, although further in-detail analytical studies are required.

The mineral alteration styles identified throughout the WSB represent important vectors for both IOA- and IOCG-deposits in northern Norrbotten (e.g. Martinsson et al., 2016) and world-wide (e.g. Corrieveau and Mumin, 2012). In our study, alteration styles typical for comparable IOA-IOCG districts elsewhere (i.e. calcic-sodic and potassic ± ferroan assemblages) are consistently developed along the WSB and show a spatial association with certain generations of structures that can be correlated with IOA- and/or IOCG-style mineralization (e.g. the Ekströmsberg and Tjårrojåkka areas).

### 6.4.1 Summary of hydrothermal alterations, metamorphism, and its relation to deformation

To summarize the timing of the dominant alteration styles the following scheme is presented (Fig. 16): Peak metamorphism is indicated by hornblende + epidote + albite implying upper greenshist to lower amphibolite facies conditions around the Ekströmsberg area. The timing of peak metamorphism is interpreted as early to syn-$D_1$.

$D_1$ is related to the following hydrothermal/metasomatic mineral assemblages:

- Regional conformable albite-scapolite alteration or growth of albite-scapolite porphyroblasts.
- Regional pervasive amphibole + magnetite alteration.
- Discordant vein fill amphibole + magnetite + albite alteration.
- Shear zone hosted pervasive actinolite + tremolite alteration.

$D_2$ is related to the following mineral alteration assemblages:

- Regional selectively pervasive K-feldspar alteration.
- Local discordant vein fill K-feldspar + epidote + iron oxide + sulphide alteration.
- Local discordant vein/shear band fill scapolite alteration.
- Local shear band fill of biotite + magnetite + K-feldspar alteration.
- Local fracture-fill epidote alteration.
- Retrograde sericite alteration.

### 7 Conclusions

Based on new structural mapping and microstructural investigations presented in this study, two major compressional events affecting the WSB have been identified. These deformation events, $D_1$ and $D_2$, developed a pronounced structural signature



and can be correlated with different types of mineral alteration assemblages. Early $D_1$ produced a steep SW- to WSW-dipping heterogeneously developed penetrative and continuous $S_1$ foliation related to $F_1$ folds with either tight east-verging symmetries with a shallow NW-plunge or tight upright symmetries with steep plunges. The steep $F_1$-plunges are in this paper interpreted to be a result of later $D_2$-transposition. Shear zones recording reverse oblique west-side-up kinematics were developed during

$D_1$ giving rise to an undulating NNW-trending configuration of magnetic lineaments.

The $S_1$ foliation was folded during $D_2$ into near-cylindrical $F_2$ folds with steep N- and S-plunges. Axial plane parallel $S_2$ foliation is rarely developed in relation to these folds and when present, $S_2$ is spaced. The inability of $D_2$ to produce a tectonic foliation over large areas indicates that the conditions during $D_2$ were of low-pressure type. Instead, the finite $D_2$ strain was partitioned into pre-existing shear zones, lithological contrasts, and rheologically favorable lithotypes, reactivating these

structures with dip-slip west-side-up sense-of-shear. Synchronously, E-W trending sinistral strike-slip shear zones were active partly displacing the NNW-grain.

$D_1$ is associated with regional scapolite $\pm$ albite alteration formed coeval with regional magnetite $\pm$ amphibole alteration and calcite under epidote-amphibolite metamorphism. The hydrothermal alteration linked to the $D_2$ deformation phase is more potassic in character and dominated by K-feldspar $\pm$ epidote $\pm$ quartz $\pm$ biotite $\pm$ magnetite $\pm$ sericite $\pm$ sulphides, and calcite.

$D_2$ is associated to regional selectively pervasive K-feldspar alteration (replacing albite) and retrograde sericite and epidote but $D_2$-alteration is to a large extent characterized by being hosted by structures. This implies that our field based observations support an early timing of calcic-sodic alteration whereas a late timing is interpreted for potassic $\pm$ ferran style of alteration assemblages and the time difference may have been 80 m.y. based on geochronological circumstantials reported from northern Norrbotten.

*Data availability.* Structural field measurements and analyzed thin sections are available from the corresponding author.

*Author contributions.* JBHA performed the geological mappings within the areas Eustiljåkk and eastern part of the Kaitum West. JBHA and TEB performed the mappings of the Ekströmsberg area and the areas in-between Ekströmsberg and

Eustiljåkk. TEB and EPL performed the mappings within the areas Fjällåsen-Allavaara and Tjårrojåkka. JBHA, TEB, and EPL did the mappings together of the western parts of the Kaitum West area and the areas in-between Kaitum West and Ekströmsberg. JBHA performed the structural analysis of Eustiljåkk, Ekströmsberg and Kaitum West. TEB performed the structural analysis of the Tjårrojåkka and Fjällåsen-Allavaara areas. All microstructural analysis used in this paper was performed by JBHA. The writing was performed by JBHA with much help from EPL and TEB. Contributions are as follows:

JBHA (50%), TEB (25%), EPL (25%).

*Competing interests.* The authors declare no conflict of interest.



*Acknowledgements.* This study is financed by the Centre of Advanced Mining and Metallurgy (CAMM), who is thanked for its financial contributions. Parts of this work were undertaken as part of the VINNOVA project "Multi-scale 4-dimensional geological modelling of the Gällivare area" and SGU's "Barents project" in northern Norrbotten. Thorkild Maack Rasmussen is thanked for the processesing of the magnetic data and for compiling the magnetic maps. Hugo Hedin Baastrup is thanked

for his field assistance in the eastern part of the Kaitum West and Eustiljåkk key areas. Software from Midland Valley was used for data collection and subsequent structural analysis.

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





**Figure 1:** Generalized geology of northern Norrbotten highlighting Palaeoproterozoic metasupracrustal belts. Simplified and modified after Bergman et al. (2001).

**Figure 2:** Geological map of the Western supracrustal belt showing the key areas in relation to the belt. Lithological contacts simplified and modified after Offerberg (1967), Witschard (1975), and Bergman et al. (2001). Coordinates: Sweref99.

**Figure 3:** First vertical derivative aeromagnetic map of the Western supra crustal belt overlaid by first vertical derivative ground-magnetic map of Ekströmsberg (in Ekströmsberg). Outline of the key areas, observation points and structures are the same as in Figure 2. Red colour: magnetic high, Blue colour: Magnetic low. Magnetic data from Frietsch et al. (1974) and Bergman et al. (2001). Coordinates: Sweref99.

**Figure 4:** Geological map of the Eustiljåkk key area. Modified after Offerberg (1967). Coordinates: Sweref99.

**Figure 5:** Stereographic equal area projections highlighting A) Low strain $S_1$. B) High strain S-fabrics. C) Stretching and mineral lineations

**Figure 6:** Characteristics of Eustiljåkk key area. A) Low intensity $S_1$-foliation, X688743 Y7542711. B) Parasitic $F_1$ folding of metasedimentary horizons, X687043 Y7539378. C) Scapolite altered mafic dyke associated to N-S directed high strain zones, X687505 Y7543037. Coordinates: Sweref99.

**Figure 7:** Geological map of the Ekströmsberg key area. Modified after Offerberg (1967). Coordinates: Sweref99.

**Figure 8:** Field images and thin section photographs of characteristics of the Ekströmsberg area. A) Actinolite-tremolite L-tectonite overprinted by calcite alteration, X689886 Y7530423. B) Semi-conformable scapolite replacement of magmatic bedding in Rhyacian basalt, X688163 Y7530345 C) Magmatic bedding in a felsic volcanic rock resembling a weak tectonic cleavage in outcrop, X691591 Y7527763. D) Micrograph of the outcrop in Fig. 8C. E) Micrograph of crenulation from same locality as Fig. 8B, X688163 Y7530345.
F) SCC′ fabric along north plunging stretching lineation in the NNW directed grain, X688714 Y7530350. G) SC fabric′ along near vertical stretching lineation in the NNW directed grain, X688167, Y7530354. H) SC fabric along shallow east plunging stretching lineation along the E-W directed grain, X690276 Y7527251. I) Brittle feldspar along the E-W directed grain, same location as Fig. 8I. Coordinates: Sweref99

**Figure 9:** Geological map of the Tjårrojåkka key area. Modified after Offerberg (1967). Coordinates: Sweref99.

**Figure 10:** Field images of key localities in the Tjårrojåkka key area. A) Isoclinal $F_1$ folding, X675916 Y7515915. B) Open chevron-style $F_2$ folding with spaced $S_2$, X676285 Y7516107. C) Open concentric $F_2$ folding with spaced $S_2$, X675985 Y7516176. Coordinates: Sweref99

**Figure 11:** Geological map of the Kaitum West key area. Modified after Offerberg (1967). Coordinates: Sweref99.

**Figure 12:** Thin section and field images/sketches of key localities in the Kaitum West key area. A) Micrograph of $S_0/S_1$ composite
fabric in volcanosedimentary rock, X698952 Y7505460. B) Isoclinal mesoscale folding of the $S_0/S_1$ fabric Fig. 12 A. C) Volcanosedimentary unit showing a high strain cleavage sub-parallel to $S_0$. The $S_0/S_1$ fabric is transposed into the direction of $D_2$ shear bands, X700402 Y7504830. D) Simplified sketch of Fig. 12 C. E) SC fabric and asymmetric dextral sigma-sigmoid viewed along shallow south-plunging stretching lineation, X702488 Y7504983. F) SC fabric indicating sinistral kinematics viewed along steep north-northwest-plunging stretching lineation, X693703 Y7519820. Coordinates: Sweref99

**Figure 13:** Geological map of the Fjällåsen-Allavaara key area. Modified after Witschard (1975). Coordinates: Sweref99.

**Figure 14:** Field images of characteristics of the Fjällåsen-Allavaara key area. A) High intensity foliation, X721253 Y7473543. B) Isoclinally folded quartz and amphibole veins with related axial planar $S_1$ cleavage. Brittle-ductile $S_2$ with dextral sense-of-shear, X721192 Y7473733. C) Tigh $F_1$ folding of $S_0$, X714642 Y7479559. D) Asymmetric lithic sigma clast viewed along steep north plunging stretching lineation, X715483 Y7478182 E) Isoclinal $F_1$ gently refolded by $F_2$, X713553 Y7480026. F) Gentle $F_2$ folding of $S_0/S_1$ with
associated brittle-ductile $S_2$, X721194 Y7473733. Coordinates: Sweref99.





**Figure 15: Field- and thin section images of alteration styles throughout the WSB. A) Scapolite porhyroblasts, X696049 Y7507684 B) Scapolite in veins and patches transposed by later shear bands, X697177 Y7532766. C) Scapolite + albite overprinting magnetite + amphibole, X695488 Y7507194. D) Vein-hosted magnetite + amphibole with reddish albite haloes, X695459 Y7507412. E) Calcite overprinting actinolite-tremolite L-tectonite in Fig. 8A. Calcite aligned with $S_1/S_2$ with undeformed granular epidote at grain**
5    **boundaries, X689886 Y7530423. F) Reddish calcite overprinting ductile shear zone fabrics, X698369 Y7534383. G) Albite + hornblende + epidote metamorphic fabric aligned axial-planar parallel $S_1$. Folded vein-hosted hornblende + epidote, X690267 Y7529944. H) K-feldspar + epidote + Fe-oxide + sulphide + malachite overprinting pervasive magnetite + amphibole, X694578 Y7506821. I) Selectively pervasive K-feldspar alteration accompanied by epidote, X696619 Y7508353. J) Selectively pervasive K-feldspar replacing albite in same outcrop as Fig. 8C, X691491 Y7527635. K) E-W directed $D_2$-shear zone with magnetite bands,**
10    **X690231 Y7527374. L) Shear band hosted biotite + magnetite + quartz + K-feldspar + muscovite from the locality in Fig. 15K.**

**Figure 16: Summary of mineral alteration assemblages in the WSB and their inferred timings.**

**Figure 17: Lower hemisphere equal area stereographic projections of lineation throughout the WSB. Cones represent 30° circles. A) $L_1$ Stretching- and mineral lineation measured on $S_1$ foliation planes in low strain blocks. B) Stretching and mineral lineation measured in high strain zones.**

15    **Figure 18: Conceptual model of the structural development of the WSB.**





**FIGURE 1**





**FIGURE 2**



**FIGURE 3**







**FIGURE 4**



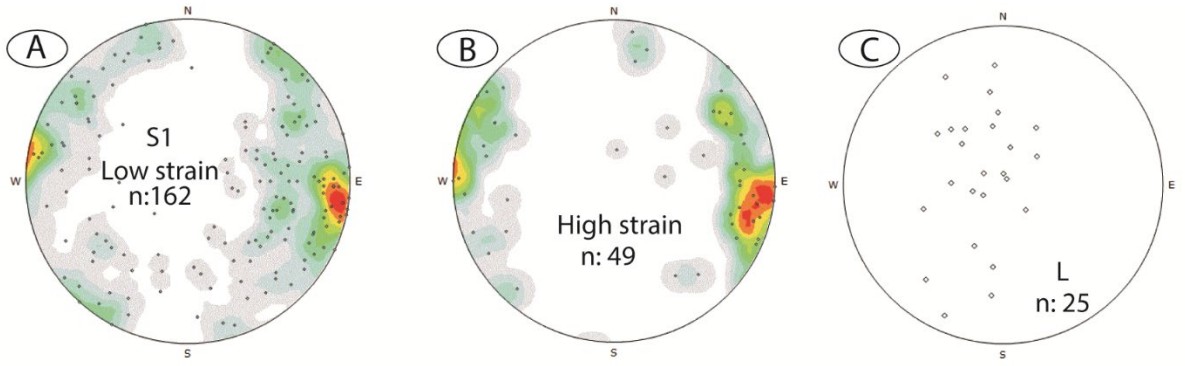

**FIGURE 5**



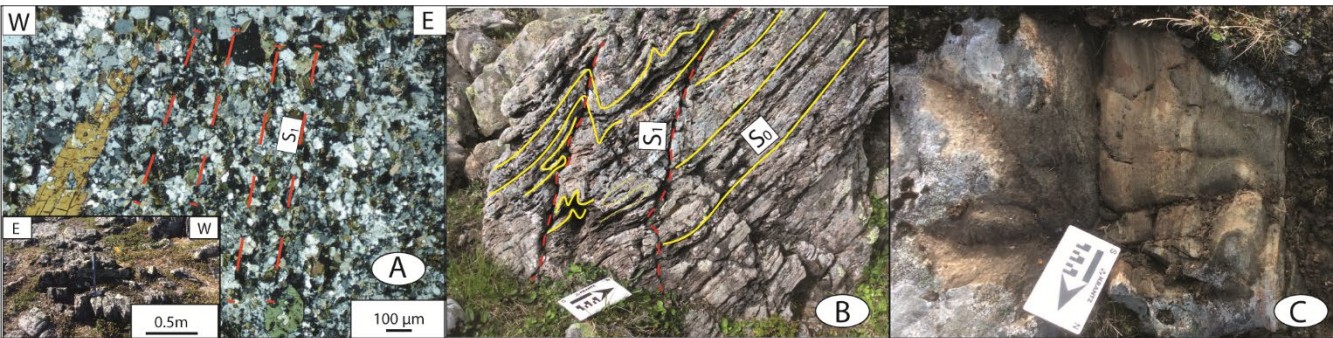

**FIGURE 6**





**FIGURE 7**





**FIGURE 8**





**FIGURE 9**



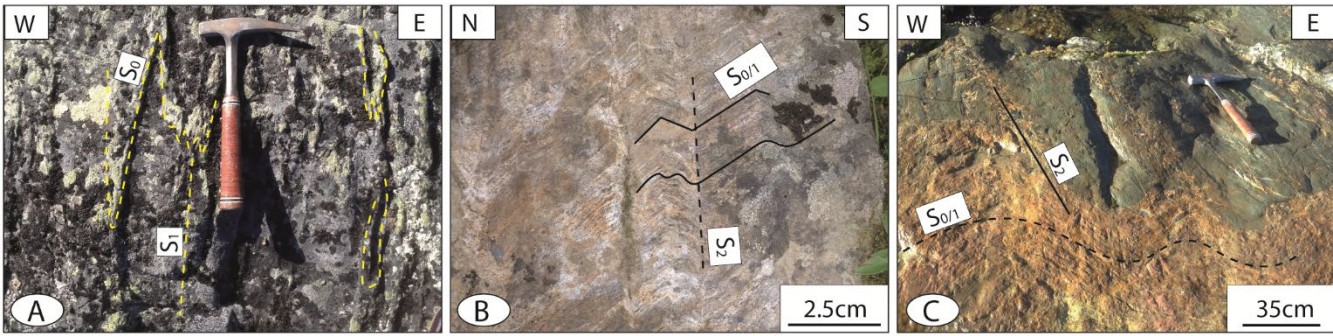

**FIGURE 10**





**FIGURE 11**



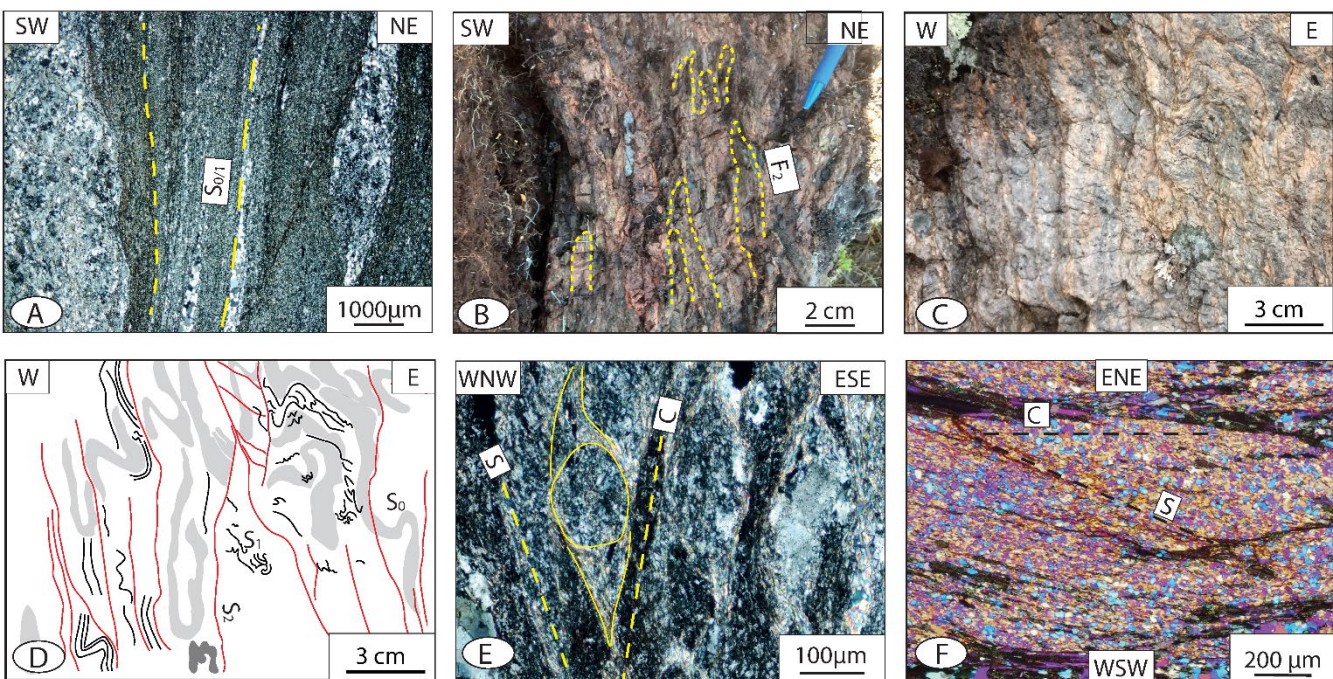

**FIGURE 12**



**FIGURE 13**



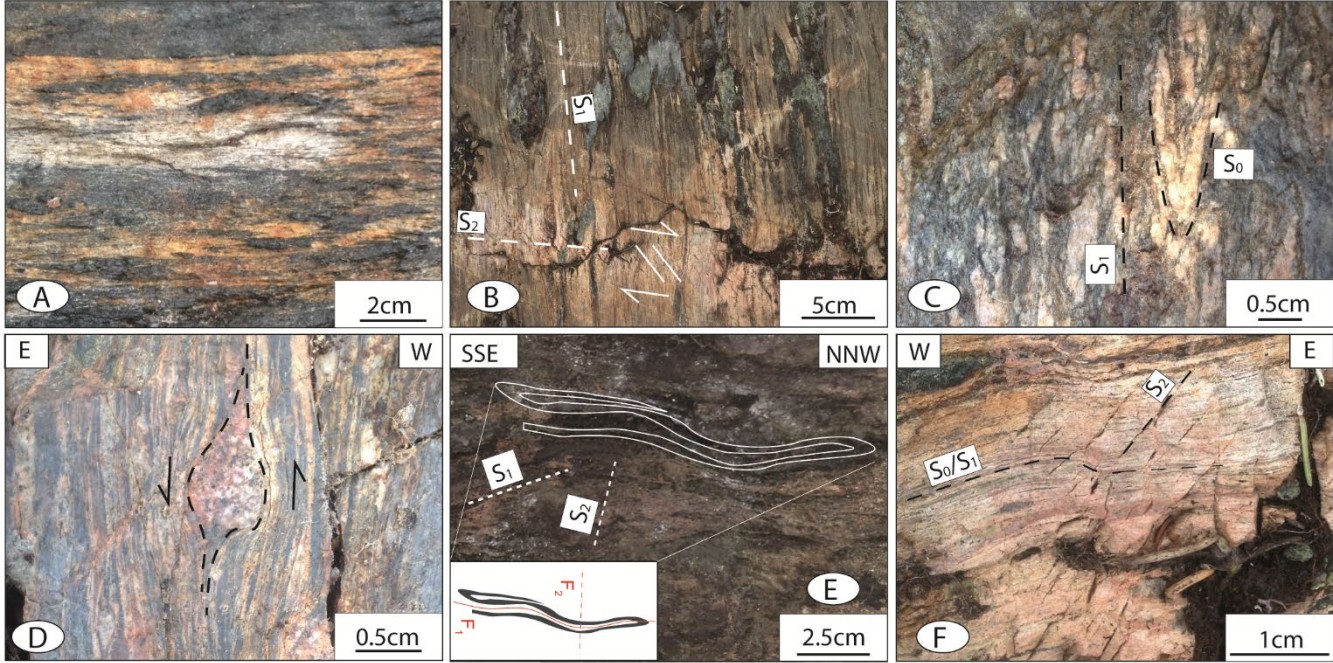

**FIGURE 14**



**FIGURE 15**





| Mineral assemblage | Texture | Distribution | Intensity | Timing |
|---|---|---|---|---|
| Scp+Ab | Semi-conformable | Regional? | Weak-strong | Pre-tectonic |
| Scp+Ab | Porphyroblastic | Regional | Weak-strong | Syn to post-$D_1$ |
| Scp+Ab | Selective-pervasive dissiminated | Regiona? | Weak-strong | $D_1$ |
| Scp+Ab | Vein haloes | Local | Weak-strong | $D_1$ |
| Scp+Ab | Selective shear band haloes | Local | Weak-strong | $D_2$ |
| Amp+Mag | Pervasive disiminated | Regional | Weak-strong | $D_1$ |
| Amp+Mag+Ab | Massive vein infill | Local | Strong | $D_1$ |
| Hbl+Epi+Ab | Pervasive dissiminated | Regional metamorphic | Moderate | Pre to syn-$D_1$ |
| Akt+Tre | Pervasive dissiminated | Local shear zone hosted | Strong | $D_1$ |
| Cal | Vein hosted | Regional | Weak-strong | $D_1$ |
| Cal | Pervasive | Local shear zone hosted | Strong | $D_2$ |
| Kfs | Selective consumption of albite | Regional | Weak-strong | $D_2$ |
| Kfs+Epi+Sul | Selective pervasive | Local? | Intense | $D_2$ |
| Kfs+Epi+Qtz | Selective pervasive | Local? | Weak-strong | $D_2$ |
| Bt+Mag+Kfs | Selective pervasive | Shearband hosted | Strong | $D_2$ |
| Epi | Selective pervasive | Local? | Strong | $D_2$ |
| Epi | Patches on fracture planes | Local? | Weak-intense | Syn to post-$D_2$ |
| Ser | Selectively consuming Kfs | Regional | Weak-moderate | Syn to post-$D_2$ |

**FIGURE 16**





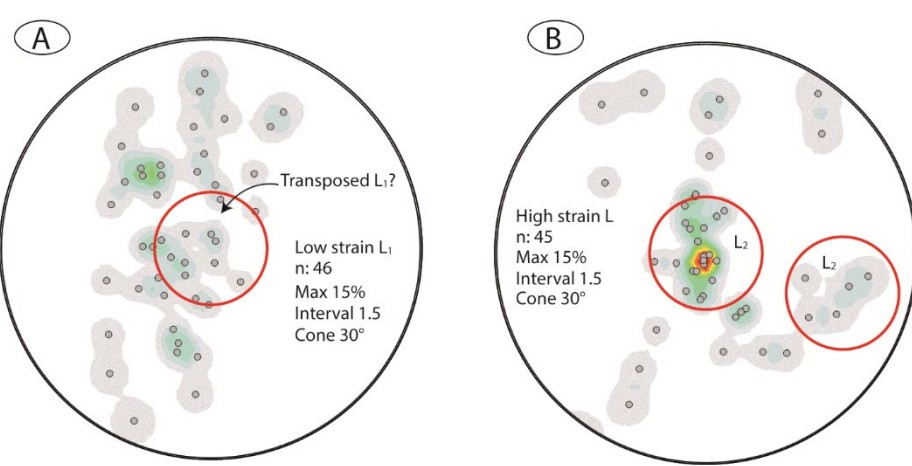

**FIGURE 17**





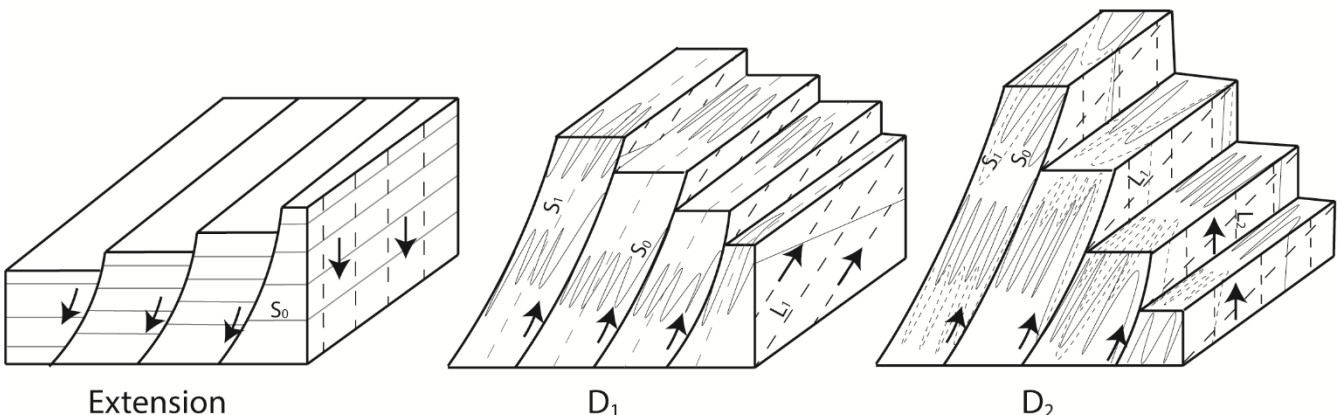

**FIGURE 18**