# Peer review of "Evolution of structures and hydrothermal alteration in a Paleoproterozoic supracrustal belt: Constraining paired deformationfluid flow events in an Fe and Cu-Au prospective terrain in northern Sweden"

_Solid Earth, 2019_

## Referee Comment (RC1) · Kunfeng Qiu (Referee) · 26 Dec 2019

Review of SE-2019-150: "Evolution of structures and hydrothermal alteration in a Palaeoproterozoic metasupracrustal belt: Constraining paired deformation-fluid flow events in a Fe and Cu-Au prospective terrain in northern Sweden" by Andersson et al., This manuscript presents systematic field-based investigation of an Orosirian meta-supracrustal belt, including geological mapping and petrographic and paragenetic as-

sessment. The authors constrain structural evolution and furthermore link hydrothermal events to specific phases of deformation. In my opinion, the presented descriptions support the aforementioned conclusions, and this makes the manuscript an interesting contribution to international community, so presumably suitable for acceptance. However the manuscript presented for reviewing has some issues and I therefore recommend minor correction before its publication. Major comments: 1) The authors have constrained the timing of deformation by previous geochronological data. In my opinion, the paper will be more attractive if the authors could add some basic geologic characteristics, geochronology, and time scale of mineralization in the figure 16 and section 6.4.1

General comments: 1) Please unify the labels of the figures (uppercase or lowercase) in text and figure and explain the abbreviations. 2) Figure 6: Add some spaces between the images 3) Figure 8d: Delete the extra box. 4) I suggest the authors add the location (area) in the captain of figure 15. It is difficult to link the coordinates to the different key areas. 5) In section 6.4, the author mentioned hornblende + epidote + plagioclase assemblage in figure 15e. But there is no corresponding information in the figure.

Kunfeng Qiu, China University of Geosciences, Beijing

———————————————

---

## Referee Comment (RC2) · Jochen Kolb (Referee) · 1 Jan 2020

Dear Florian,

please find enclosed my review of the manuscript entitled "Evolution of structures and hydrothermal alteration in a Palaeoproterozoic metasupracrustal belt: Constraining paired deformation-fluid flow events in a Fe and Cu-Au prospective terrain in northern Sweden" by Andersson, Bauer and Lynch. The authors use mapping, structural

analysis, microstructures and mineralogy in order to describe deformation and related hydrothermal alteration systems in their study area. They interpret the structural data and conclude that the structures are best explained by two subsequent deformation stages with different stress field. They try to relate hydrothermal alteration assemblages to the different deformation stages using field observations and microstructural data. The observational data is of very good quality and the structural interpretation is valid. The manuscript is, however, poorly written and the interpretation of the mineralogical and microstructural data needs major improvement. I recommend major revisions of the manuscript before publication. My main concerns are: 1. The wording of the manuscript is poor. The authors are imprecise and don't use the language of our science strictly. They mix up terms and use language that makes understanding of their descriptions difficult or impossible. The entire manuscript needs careful rewording and possibly the care of a native speaker. 2. The authors need to reword the entire manuscript and need to follow the two principles of writing a geology manuscript: (1) old structures or rocks need to be described before their younger counterparts; and (2) data needs to be presented and described first, interpretation follows. 3. The metamorphic and hydrothermal assemblages need to be described in much more detail. What is their relationship to foliations and lineations? The mineralogy of hydrothermal alteration zones depends on P, T, X and physicochemical parameter. This results in the situation that hydrothermal mineral assemblages may not only vary on relative timing in the geological evolution and along a PT path. They also vary with host rock composition, fluid composition, distance from the main fluid conduit etc. This causes in many situations complex hydrothermal alteration patterns and zoning in hydrothermal ore deposits. This is well-described in many similar systems elsewhere in the world. The authors need to be more careful with their petrological data and must observe and interpret with much more detail. It would help the reader, if the authors could add hydrothermal alteration zones to their lithological and structural maps.

Detailed comments: Palaeoproterozoic: The stratigraphic commission has changed the general way of spelling this into Paleoproterozoic (also Archean, etc.). Sulphide:

**SED**

The now generally accepted spelling of this word in economic geology papers is "sulfide". Title: Delete "meta" and use supracrustal belt. I personally prefer "greenstone belt", because often not all of the rocks contained in such a belt are strictly supracrustal (you also describe dykes for example). Mineralization: "Mineralization" is a process not a thing. Check your wording accordingly. Introduction: There is a lot, which is repeated and detailed in later chapters. The introduction should introduce the problem and specify the research questions and the approach. This is only partly true here. Why is the study important? What will the addition to science be? Why is relating hydrothermal alteration to structures important? How is the situation elsewhere in similar terranes with IOCG deposits, Canada, Brazil, Australia (e.g. Tennant Creek, Mount Isa), Mauritania. . . . . ..? Regional Geology: This chapter is poorly worded and poorly structured. The data and observation must be presented before interpretation. A clear stratigraphy is necessary. I suggest preparing a table or a sketch to help the reader. In the text, the nomenclature has to be used strictly and consistently. Methods: No Leapfrog model is shown in this paper – adjust this chapter to the methods used for generation of the data presented in this manuscript. Results: Describe your data from old to young. This is a geology paradigm that makes sense, because old structures are always overprinted by young structures. Don't shift between scales. Make a description at regional, district, local, outcrop, sample, thin section scale and organise accordingly. Chapter 6.4.1 is needed much earlier in the manuscript. Figures: Correct legend and check for completeness of all legends in all maps. Add legend to the stereographic plots. Page 1 Line 16 ff.: Relationship between alteration assemblages is unclear. Why are there two regional hydrothermal alteration assemblages? What is their relationship? What is the importance of calcite? This needs explanation that is more careful and rewording. Line 20: Avoid the term "brittle-ductile". This term is derived from geophysical (seismic) investigation of the Earth's crust and defined as a zone of velocity change of seismic waves. It has no geological meaning. I can show you examples of brittle-ductile behaviour of rocks at 250°C and at 650°C. I suggest avoiding the term, because it does not add any information. Page 2 Line 1: This sentence is misleading as is the referencing. The two papers do not describe the situation in Norbotten as your wording indicates. It is unclear to me, how metamorphic rocks with only restricted porosity (and generally no permeability) can focus fluid flow. Fluid follows the hydraulic gradient and can only migrate through permeable rock. Line 3 ff.: This is repeated from the previous page. Reword and avoid repetition. Line 12: There is a problem here and also elsewhere in the text with the terminology. You define "Svekokarelian" as an orogeny. Here you use it as a stratigraphic term. This is confusing for the reader and needs to be avoided. In the regional geology chapter, a description of stratigraphy (old to young) and metamorphism and deformation (if information is available) is needed. The terminology needs to be clearly defined (if helpful with a table) and strictly used in the entire manuscript. Page 4 Line 25 ff.: What was used to constrain the PT conditions? The method or mineral assemblage needs to be stated. Otherwise, the reader cannot evaluate the quality of the data himself. Is it the metamorphic peak that is recorded? What is the approximate timing of metamorphism? You state that PT conditions raised from greenschist to amphibolite facies regionally, but then outline granulite facies conditions. What is true? Line 30 ff.: You say that the deformation is polyphase, but you only describe one stage of deformation in the text. Page 10 Line 6: This needs much more explanation. How does dip-slip relate to compressional deformation? Generally, compressional deformation is related to folding, strike-slip and reverse shearing and not normal (dip-slip) deformation. Line 10 ff.: You need to organize. E.g. you describe brittle and ductile deformation of feldspar, then describe many other things and then describe ductile deformation of quartz. See above; organize the structural description according to scale, either from small-scale to regional-scale or vice versa. The quartz fabrics (D1/D2 similar, low-T) contradict the feldspar fabrics (D1 >450°C, D2 low-T) – why? Page 12 Line 15 ff.: This remains unclear. If scapolite forms porphyroblasts, then it at least postdates D1 (otherwise, it must be porphyroclastic). If the veins are deformed by D2, then they predate D2. Which time constraints do you have between D1 and D2 to say that the veins postdate the scapolite porphyroblasts? Why is scapolite not formed at the same time? Page

13 Line 4 ff.: This is very confusing. You jump in your description from early to young and vice versa. The reader is unable to follow this. The important message is, which hydrothermal alteration assemblage(!) is temporally related to which structural fabric and to which stage in the metamorphic PT evolution (prograde, peak, retrograde). You also need to distinguish between hydrothermal alteration and metamorphic mineral assemblages. They form via completely different processes. This needs careful rewording. Page 14 Line 20 ff.: What about the feldspar microstructures described above? What do they indicate? What about the mineral assemblages? Which PT conditions do they indicate? This needs much more discussion and integration of microstructural and petrological data. Likely much better PT constrains are possible by such an integration. Page 15 Line 5: Relatively low P and upper crustal conditions are also true for D1 deformation. This does not add any information and is a very vague and imprecise statement. See also comment above. Line 6 ff.: This needs more discussion. It remains completely unclear, why you add information from other localities and which relevance that may have. You first need to constrain PT conditions and relative timing of deformation, metamorphic and hydrothermal stages in your study area before you can compare those to other (similar or connected) areas. Readers, who are not familiar with the regional geology, will be confused and cannot decide on the relevance of the data presented here for comparison. Line 13 ff.: This part suffers from bad wording and poor organization. Please reword and discuss from old (L1) to young (L2). The mineral species that define the stretching lineation may help in constraining relative timing and PT conditions. Page 16 Line 20 ff.: Why are you presenting this? There is a lot of speculation and the reader gets no idea, what the relevance of this is: delete. Line 26 ff.: I am confused: Why do D2 shear bands indicate D1 deformation? Page 17 Line 2 ff.: What is the timing of metamorphism? The epidote-amphibolite facies in pillow basalts can be very early in the evolution and may represent seafloor metasomatism. Alternatively, a similar mineral assemblage can form during regional metamorphism. This needs more discussion of evidence of relative timing. Page 19 Line 4: If the scapolite is really porphyroblastic everywhere, then it must (by definition)

be post-tectonic. You say that the scapolite porphyroblasts occur in low-strain areas, which may mean that scapolite growth postdates D1 here but is pre-, syn- or post-D1 in the high-strain zones. This needs more discussion and more precise investigation of the relative timing. Furthermore, you contradict yourself with the interpretation of the absolute age data. You say that D1 is 1.88-1.86 Ga, that scapolite alteration is syn- to late-D1 and that scapolite formed (together with titanite) at 1.9 Ga. This is 20-40 m.y. earlier than D1 and not syn- to late-D1. Moreover, you do not provide the precision of the geochronological data, which makes further evaluation by the reader impossible. Is scapolite formed during seafloor alteration at 1.9 Ga? Line 10 ff.: The discussion about Cl/Br geochemistry and fluid source remains unclear. If you have two generations of scapolite, what will whole rock data tell you? Why do you consider an evaporitic source, when the data contradicts this? This needs much more discussion, if this is important for the conclusions of this work. I am not sure, where this will lead to?? You don't really have shown constrains on the relative timing of scapolite alteration. This makes it really difficult to follow. Line 17 ff.: If you would follow the geological principle of describing the oldest features first, your text would be much easier to follow. You get not all possible information out of your data. There is much more on P-T-D-X relationships in your data that needs to be presented and discussed. The important part is that hydrothermal alteration assemblages are controlled by complex P-T-X relationships. The mineral presence in the hydrothermal alteration assemblages depends not only on fluid composition, but also on host rock composition and fluid rock ratio. Thus, it is very common in hydrothermally altered regions to find different hydrothermal alteration assemblages in different host rocks that formed at the same time or to find zoning of hydrothermal alteration assemblages depending on the distance from the main hydrothermal fluid conduit (e.g. a shear zone or a pluton). All this needs to be worked out from your data and displayed in maps and discussed accordingly.

Kind regards Jochen Kolb

Please also note the supplement to this comment:

[Figure]

https://www.solid-earth-discuss.net/se-2019-150/se-2019-150-RC2-supplement.pdf

**Supplement:**

[revised manuscript text omitted]

Rhyacian greenstones (tuffites and basalts)

Orosiran c. 1.89 Ga mafic-intermediate metavolcanic/volcanosedimentary rocks (basaltic to andesitic)

Orosirian c. 1.89 Ga felsic metavolcanic/volcanosedimentary rocks (dacitic to rhyodacitic)

Orosirian metasedimentary rocks (Conglomerates, arenites)

Orosirian 1.88-1.86 Ga plutonic rocks

Orosirian c. 1.80 Ga plutonic rocks

Ductile high strain zones

Brittle-ductile high strain zones

Observation points

Fe

Cu

References:
Offerberg (1967), Witschard (1975), Bergman et al. (2001)

[Figure]

[Figure]

[Figure]

[Figure]

[Figure]

[Figure]

[Figure]

[Figure]

[Figure]

[Figure]

[Figure]

[Figure]

[Figure]

[Figure]

| Mineral assemblage | Texture | Distribution | Intensity | Timing |
|---|---|---|---|---|
| Scp+Ab | Semi-conformable | Regional? | Weak-strong | Pre-tectonic |
| Scp+Ab | Porphyroblastic | Regional | Weak-strong | Syn to post-$D_1$ |
| Scp+Ab | Selective-pervasive dissiminated | Regiona? | Weak-strong | $D_1$ |
| Scp+Ab | Vein haloes | Local | Weak-strong | $D_1$ |
| Scp+Ab | Selective shear band haloes | Local | Weak-strong | $D_2$ |
| Amp+Mag | Pervasive disiminated | Regional | Weak-strong | $D_1$ |
| Amp+Mag+Ab | Massive vein infill | Local | Strong | $D_1$ |
| Hbl+Epi+Ab | Pervasive dissiminated | Regional metamorphic | Moderate | Pre to syn-$D_1$ |
| Akt+Tre | Pervasive dissiminated | Local shear zone hosted | Strong | $D_1$ |
| Cal | Vein hosted | Regional | Weak-strong | $D_1$ |
| Cal | Pervasive | Local shear zone hosted | Strong | $D_2$ |
| Kfs | Selective consumption of albite | Regional | Weak-strong | $D_2$ |
| Kfs+Epi+Sul | Selective pervasive | Local? | Intense | $D_2$ |
| Kfs+Epi+Qtz | Selective pervasive | Local? | Weak-strong | $D_2$ |
| Bt+Mag+Kfs | Selective pervasive | Shearband hosted | Strong | $D_2$ |
| Epi | Selective pervasive | Local? | Strong | $D_2$ |
| Epi | Patches on fracture planes | Local? | Weak-intense | Syn to post-$D_2$ |
| Ser | Selectively consuming Kfs | Regional | Weak-moderate | Syn to post-$D_2$ |

[Figure]

[Figure]

Extension          D₁          D₂

---

## Author Comment (AC3) · 27 Jan 2020

[revised manuscript text omitted]

Rhyacian greenstones (tuffites and basalts)

Orosiran c. 1.89 Ga mafic-intermediate metavolcanic/volcanosedimentary rocks (basaltic to andesitic)

Orosirian c. 1.89 Ga felsic metavolcanic/volcanosedimentary rocks (dacitic to rhyodacitic)

Orosirian metasedimentary rocks (Conglomerates, arenites)

Orosirian 1.88-1.86 Ga plutonic rocks

Orosirian c. 1.80 Ga plutonic rocks

Ductile high strain zones

Brittle-ductile high strain zones

Observation points

Fe

Cu

References:
Offerberg (1967), Witschard (1975), Bergman et al. (2001)

10 km

Eustiljåkk, Fig. 4

Ekströmsberg, Fig. 7

Tjårrojåkka, Fig. 9

Kaitum West, Fig. 11

Fjällåsen-Allavaara, Fig. 13

Viscaria Cu

Kiruna

Vieto

Fjällåsen

698203

723203

7536582

7511582

7486582

[Figure]

Eustiljåkk, Fig. 4

Vieto

Kiruna

Viscaria Cu

Ekströmsberg, Fig. 7

Tjårrojåkka, Fig. 9

Kaitum West, Fig. 11

Fjällåsen

Fjällåsen-Allavaara, Fig. 13

698203

723203

7536582

7511582

7486582

10 km

n = 25
Interval size = 2.52
max. 12 %

L

n = 10

Fjällåsen Cu

n = 41
Interval size = 2.7
max. 21 %

L
n = 11

Allavaara

2 km

| | Orosirian sedimentary and volcanosedimentary rock |
| | Orosiran intermediate metavolcanic/volcanosedimentary rocks (basaltic-andesitic) |
| | Orosirian felsic metavolcanic/volcanosedimentary rocks (dacitic to rhyodacitic) |
| | Plutonic rocks PMS suite (gabbroids) |
| | Plutonic rocks Lina suite (granitoids) |

Reverse ductile shear zone

Semi-brittle high strain zones

Shear sense

Bedding

Cleavage: inclined, vertical

High strain ceavage: inclined, vertical

Lineation: plunging, vertical

Fold axis: plunge in degrees

Stratigraphic way up

Synform, with plunge

Sulfide ore prospect

[Figure]

[Figure]

[Figure]

| Mineral assemblage | Texture | Distribution | Intensity | Timing |
|---|---|---|---|---|
| Scp+Ab | Semi-conformable | Regional? | Weak-strong | Pre-tectonic |
| Scp+Ab | Porphyroblastic | Regional | Weak-strong | Syn to post-D$_1$ |
| Scp+Ab | Selective-pervasive dissiminated | Regiona? | Weak-strong | D$_1$ |
| Scp+Ab | Vein haloes | Local | Weak-strong | D$_1$ |
| Scp+Ab | Selective shear band haloes | Local | Weak-strong | D$_2$ |
| Amp+Mag | Pervasive disiminated | Regional | Weak-strong | D$_1$ |
| Amp+Mag+Ab | Massive vein infill | Local | Strong | D$_1$ |
| Hbl+Epi+Ab | Pervasive dissiminated | Regional metamorphic | Moderate | Pre to syn-D$_1$ |
| Akt+Tre | Pervasive dissiminated | Local shear zone hosted | Strong | D$_1$ |
| Cal | Vein hosted | Regional | Weak-strong | D$_1$ |
| Cal | Pervasive | Local shear zone hosted | Strong | D$_2$ |
| Kfs | Selective consumption of albite | Regional | Weak-strong | D$_2$ |
| Kfs+Epi+Sul | Selective pervasive | Local? | Intense | D$_2$ |
| Kfs+Epi+Qtz | Selective pervasive | Local? | Weak-strong | D$_2$ |
| Bt+Mag+Kfs | Selective pervasive | Shearband hosted | Strong | D$_2$ |
| Epi | Selective pervasive | Local? | Strong | D$_2$ |
| Epi | Patches on fracture planes | Local? | Weak-intense | Syn to post-D$_2$ |
| Ser | Selectively consuming Kfs | Regional | Weak-moderate | Syn to post-D$_2$ |

[Figure]

[Figure]

Extension    $D_1$    $D_2$

---

## Author Response (AR1)

**List of changes se-2019-150**

Dear topical editor,

In this document we list the changes made to the manuscript "se-2019-150" in accordance with the referee comments made by Dr. Qui (RC1) and Dr. Kolb (RC2).

The changes listed below can also be found as additional comments to each referee comment in the author's response to each reviewer (attached). The numbered changes below correlates to the numbered referee comments in the authors response. For several changes, further explanation/motivation can be found in the additional comments to the referee comments in the authors response documents.

Some repetition do occur in the listed changes below but this is done in order show that we have considered each of the concerns of the reviewers.

Three new figures have been produced in order to clarify questions raised by the reviewers. Figure 4, Figure 16, and Figure 18 are new figures. Figure 4 is a summary of geological background information that we hope will help international readers. Figure 16 shows a metamorphic mineral association and its relation to deformation that was produced in order make the distinction between metamorphic and hydrothermal mineral associations clearer. Figure 18 is a summary map of hydrothermal alteration. One additional image was included in Figure 17. This image was down prioritized during the production of the first submitted manuscript, but since we could free space in Figure 17 by including Figure 16 this new image showing K-feldspar + epidote alteration could be included. We hope that this is ok since we think that the image adds information on how the potassic-ferroan alteration styles developed during D2 may appear in field.

Yours Sincerely,

*Joel Andersson,*
*Lead and corresponding author*
*PhD candidate in ore geology*
*Luleå University of Technology*
*Department of Civil, Environmental and Natural Resources Engineering*
*Division of Geosciences and Environmental Engineering*
*SE-971 87 Luleå*
*Sweden*
*Phone: +46-920493549*
*Mobile: +46-730821280*
*e-mail: joel.bh.andersson@ltu.se*

**Review 1 Dr. Qui:**

**Changes in accordance with major comments**

1. We have added a time-scale in Fig. 16 (Fig. 19 in the resubmitted manuscript) and included supracrustal rocks, intrusive rocks, and mineralization to clarify how these relate temporally to the deformation and hydrothermal mineral associations in the study area. In the summary of hydrothermal alterations, the bullet points now includes the parameters added to Fig. 19.

**Changes in accordance with general comments**

3. Labels have been unified. Abbreviations in figures have been explained in the captions.
4. Figure 7 (Figure 6 in previous version). We have added space between the images in accordance with the other figures in the manuscript.
5. We have removed the extra box in Fig. 8d (note that this is now Fig. 9d in the resubmitted manuscript).
6. We have added the area for each image in the caption of Fig. 15. (note that this is now Fig. 17).
7. We have changed the figure reference from Fig.15e to Fig.15g in section 6.4.
   (Fig. 15g is now Fig. 16a in the resubmitted manuscript).

**Review 2 Dr. Kolb**

We have changed the manuscript in accordance with the supplementary material (annotated PDF) provided by Dr. Kolb. The majority of the suggested changes were accepted. Where we have chosen not to change, this is motivated as a supplementary comment in the annotated PDF as well as motivated in the authors response.

**Changes in accordance with the main concerns**

1. Beyond the comments in the supplementary material attached by Dr. Kolb, we have reviewed the full manuscript, deleted parts of it and rewritten other parts, and reworded a number of passages where needed. Furthermore, terms are now used consistently, except the terms "phase" and "event" that are used interchangeable in the revised manuscript.
2. We have reworded passages that could be clarified by rewording throughout the entire manuscript. The text is restructured following the structure old-young except one paragraph in the Discussion on page 16 that is structured young-old. Motivation to the young-old structure of the paragraph can be found as a supplementary comment in the attached author response to RC2.
3. We have reformulated much of the text focusing on hydrothermal and metamorphic mineral associations. We have separated hydrothermal and metamorphic mineral associations into different figures (Fig. 16 and Fig. 17) and discuss these separately in the text. We now state already in the abstract that the hydrothermal mineral alterations linked to D1 may form part of one single hydrothermal system but that they are separated in this study on the basis of crosscutting relationships. This is also stated in the chapters of the paper. In the case of calcite alteration, we have clarified that it is restricted to mafic rocks and the role of carbonate alteration is discussed in the chapter Discussion. We have also produced an alteration map combining structures, aeromagnetics and alteration (Fig. 18).

**Changes in accordance with detailed comments**

1. We have changed "Palaeoproterozoic" to "Paleoproterozoic". 11 changes.
2. We have changed "sulphide" to "sulfide". 7 changes.

3. We have abandoned the prefix "meta" throughout the manuscript. Motivation to this decision is given as a last paragraph in the section 1. Introduction.

4. Where we suspect that mineralization can be read as a thing, we have changed the word "mineralization" to "ore" or "deposit" depending on context.

   Introduction. We have clarified the importance of the study and the introduction of the research gap as well as the scientific contribution.

   We have avoided repetition by deleting the second paragraph that includes a very general geological background and was originally written to "set the scene". Furthermore, background of how the study area relates to similar terrains worldwide has been added in accordance with the referee comment.

5. Regional geology: Stratigraphy together with a schematic overview of the timing of supracrustal/intrusive rocks, metamorphism, deformation, and mineralization is now included as Figure 4 and the text explains the figure.

6. Method: We have deleted the text mentioning Leapfrog in section "Methods".

7. Results: We have worked further on the text to clarify the text and made sure that the text is organized from old-young and do not mix scales.

   Regarding consistent use of time scales throughout the manuscript, Ma is now used when geochronological data is presented (e. g. U-Pb zircon age at 1902 ± 4 Ma) and Ga is used when general geological times are expressed (e.g. The Svecokarelian orogeny 1.9-1.8 Ga).

8. Chapter 6.4.1 is now extended and moved to the end of the result chapter. In the summary, the information is repeated but set into context with supracrustal/intrusive rocks and mineralization in Norrbotten. This was done in accordance with comments from Reviewer 1 Dr. Qui.

9. All legends are changed in accordance with the referee comment and the comments made on the maps in the figure captions (in the annotated PDF supplied by Dr. Kolb). We have also included a figure reference to the new alteration map (Fig. 18) in Figure 2 and 3.

10. We have added information on statistics for density plots in Figure 6 (Figure 5 in previous version).

11. Page 1, line 16ff: We have clarified that the mineral alteration associations linked to D1 may form part of one single hydrothermal system but are treated as separate in this study because they show overprinting relations. The role of calcite is clarified in the discussion where late stage carbonate deposition in IOCG-systems is discussed as well as the possibility of down-welling of meteoric fluids in the case where calcite overprints the shear zone fabric.

12. Page 1, line 20: We have changed "brittle-ductile" to "brittle-plastic" throughout the manuscript. Where the text describes that a condition is brittle-ductile, this has been changed to describe the resulting structure as brittle-plastic. In some cases where a regional feature is described, "brittle-ductile deformation zone" has been changed to "crustal-scale deformation zone".

13. Page 2, line 1: The sentence on Page 2 line 1 was deleted. The sentence before was reworded in order to keep context.

14. Page 2, line 3ff: Could not find the repetition pointed out on page 2 line 3ff.

15. Page 2, line 12: We have changed all passages where rocks are described as Svecokarelian-related. The new term is Svecofennian. The term Svecokarelian is now restricted to the orogeny.

16. Page 4, line 25ff: The role of granulite facies metamorphism has been clarified in the sentence on page 4 line 25 ff. Also, "limited PT-modelling" was changed to "microprobe data".

17. Page 4, line 30ff: The text on Page 4 line 30 ff. has been clarified for the number of deformation events in northern Norrbotten.

18. Page 10, line 6: "Reverse" is now always before "dip-slip" in the manuscript.

19. Page 10, line 10ff: The paragraph including page 10 line 10ff was reorganized into two paragraphs. The information obtained from quartz fabrics and feldspar fabrics (as well as the metamorphic textures) have been clarified with support from literature.

20. Page 12, line 15ff: The paragraph on page 12 including line 15 has been reorganized. The paragraph is now 2 paragraphs, one focusing on scapolite-albite and one paragraph focusing on magnetite-amphibole. We have clarified the interpreted timing of scapolite veining.

21. Page 14, line 20ff: In order to clarify the text and figures, we have separated hydrothermal alteration and metamorphic mineral associations to different figures (Fig. 16, Fig. 17). The figure with metamorphic minerals is now Fig. 16 and includes 2 additional micrographs showing the timing of metamorphism in relation to D1 deformation (syn-tectonic growth of hornblende). Hydrothermal mineral associations are found in Fig. 17 and includes 1 additional image Fig. 17G. We have rewritten parts of and reorganized the full section 5.6. The text now clearly separates metamorphic and hydrothermal mineral associations. It describes the alteration styles from old to young and do not concentrate on a particular mineral, but rather on the mineral associations.

22. Page 14, Line 20 ff.: The paragraph including page 14 line 20 is rewritten. We link the metamorphic mineral association to dynamic quartz recrystallization textures and discuss a possibility to why lower temperatures (400 C°) are indicated by the quartz textures in the D1 shear zones in respect to the temperatures indicated by the metamorphic texture (>450 C°). We also discuss what temperatures (400 C°) that is indicated by the quartz textures in the D2 shear zones. Also, the paragraph is restructured to describe old-young according to earlier referee comments.

23. Page 15, Line 5: We have deleted the sentence in page 15, line 5.

24. Page 15, Line 6 ff: The text on page 15 line 6ff has been clarified by stating more clear where the areas are located in Fig. 1. The importance of the comparison is motivated in the text. Based on the findings in adjacent areas together with our results from the WSB, we have clarified that a pronounced deformation during D1 followed by weaker deformation during D2 forms the deformation systematics in this part of Norrbotten.

25. Page 15, Line 13 ff.: We have reorganized the paragraph on Page 15 line 13 ff. We describe the lineation from young to old in this paragraph. The reason for this is that the L2 is rather

straight forward whereas the classification of L1 requires some degree of speculation. By describing young to old, we spare the somewhat speculative part to the end of the paragraph.

26. Page 16, Line 20 ff.: We have deleted the sentence "These fold structures could have formed in a fold-and-thrust belt (Wright, 1988; Talbot and Koyi, 1995; Angvik, 2014) or, alternatively, during basin inversion (Andersson et al. 2017)".

We have inserted a new chapter before this chapter named "6.2.1 Pre-D1-event" where we explain why our structural model takes into account that the geological setting were extensional during the deposition of the supracrustal rocks of WSB. Text is moved from the paragraph including page 16, line 20 to this new chapter.

27. Page 16, Line 26 ff. We have clarified the sentence.

28. Page 17, Line 2 ff. We have clarified the timing of the metamorphic texture by adding two new micrographs showing syntectonic growth of hornblende.

29. Page 19, Line 4: We have added a discussion on the timing of scapolitization in Norrbotten referring to earlier geochronological studies with reported precisions.

30. Page 19, Line 10 ff.: We have deleted the text discussing the Cl/Br geochemistry and fluid sources. We have reorganized the paragraph.

31. Page 19, Line 17 ff: Text rearranged old-young throughout the manuscript except for one paragraph on page 16, which is structures young-old.

The timing of metamorphism have been clarified by adding two micrographs of syntectonic growth of hornblende in the metamorphic mineral association. Also, observations of metamorphic mineral associations have been separated from observations of hydrothermal mineral associations into two separate figures (Fig. 16 and 17).

We put the PT-information obtained by the metamorphic mineral association in relation to the dynamic recrystallization quartz textures in order to obtain more PT-information out from our observations.

We have compiled a map showing hydrothermal alterations and their relation to dominant structures and magnetics, Figure 18.

Final author response to referee RC1 Dr. Kunfeng Qiu on "Evolution of structures and hydrothermal alteration in a Palaeoproterozoic metasupracrustal belt: Constraining paired deformation-fluid flow events in a Fe and Cu-Au prospective terrain in northern Sweden" by Joel B. H. Andersson et al.

Dear Dr. Qui,

We were happy to receive your review of our manuscript submitted to Solid Earth (reference no. SE-2019-150) and thank you for taking the time to read the paper and critically appraise its content. We hope that the answers provided below will satisfactorily address your comments and lead to an improved, higher quality manuscript for publication in Solid Earth.

*Joel Andersson,*
*Lead and corresponding author*
*PhD candidate in ore geology*
*Luleå University of Technology*
*Department of Civil, Environmental and Natural Resources Engineering*
*Division of Geosciences and Environmental Engineering*
*SE-971 87 Luleå*
*Sweden*
*Phone: +46-920493549*
*Mobile: +46-730821280*
*e-mail: joel.bh.andersson@ltu.se*

**Answers to major comments:**

[Figure]

**1. Comment from referee:**

The authors have constrained the timing of deformation by previous geochronological data. In my opinion, the paper will be more attractive if the authors could add some basic geologic characteristics, geochronology, and time scale of mineralization in the figure 16 and section 6.4.1

**Authors response:**

We agree that adding the suggested types of information to Figure 16 and Section 6.4.1 would be useful ideal and we consider such a holistic understanding of the regional geology as the ultimate aim. However, in order to do that in a concluding figure and text we would prefer more evidence and better regional control. At this stage of geological research in the Western Supracrustal Belt (WSB) area, we can discuss absolute time constraints and we can construct hypotheses based on our new geological data and geochronological works presented by other scholars in adjacent areas.

In order to meet this referee comment, we will need to merge our own data with results from litterature.

**Author's changes in manuscript:**

We will clarify Fig. 16 and section 6.4.1 by adding geochronological constraints on lithologies, mineralization and deformation based on available data in the literature.

**Answers to general comments:**

[Figure]

**1. Comment from referee:**

Please unify the labels of the figures (uppercase or lowercase) in text and figure and explain the abbreviations.

**Authors response:**

We will go through the manuscript and adjust accordingly.

**Author's changes in manuscript:**

Changes will be made according to the referee comment.

**2. Comment from referee:**

Figure 6: Add some spaces between the images

**Authors response:**

We agree.

**Author's changes in manuscript:**

Spaces will be added between images in Figure 6 in accordance to the other figures in the manuscript.

**3. Comment from referee:**

Figure 8d: Delete the extra box.

**Authors response:**

Definitely, thank you for pointing this out.

**Author's changes in manuscript:**

The extra box in Figure 8d will be deleted

**4. Comment from referee:**

I suggest the authors add the location (area) in the captain of figure 15. It is difficult to link the coordinates to the different key areas.

**Authors response:**

We agree that adding the areas in the captain of figure 15 will clarify figure 15.

**Author's changes in manuscript:**

In figure 15, each image will be linked to the area in which is the observation is made.

**5. Comment from referee:**

In section 6.4, the author mentioned hornblende + epidote + plagioclase assemblage in figure 15e. But there is no corresponding information in the figure.

**Authors response:**

The correct figure reference should be 15G and not 15E. We apologize for this typing error.

**Author's changes in manuscript:**

The typing error in section 6.4 will be corrected in the resubmitted manuscript.

Final author response to referee RC2 Dr. Jochen Kolb on "Evolution of structures and hydrothermal alteration in a Palaeoproterozoic metasupracrustal belt: Constraining paired deformation-fluid flow events in a Fe and Cu-Au prospective terrain in northern Sweden" by Joel B. H. Andersson et al.

Dear Dr. Kolb (JK),

We are happy to receive your detailed and thorough review on our manuscript submitted to Solid Earth (reference no. SE-2019-150). The review is detailed as well as targets questions that have implications on the overall interpretation made in this study and has the potential to significantly improve the manuscript.

We are currently reworking the manuscript. Attached is the annotated PDF-file "se-2019-150-RC2-supplement" with comments on each suggestion. Where we have preferred not to change or to change in another way than suggested, we try to motivate this in the answer to the comment in the annotated PDF. Very important to note is that this document only shows the initial reworking of the manuscript and addresses only the direct suggestions made in the PDF-file "se-2019-150-RC2-supplement". We will continue to rewrite the manuscript in accordance with the referee comments as described in the answers to the comments below.

*Joel Andersson,*
*Lead and corresponding author*
*PhD candidate in ore geology*
*Luleå University of Technology*
*Department of Civil, Environmental and Natural Resources Engineering*
*Division of Geosciences and Environmental Engineering*
*SE-971 87 Luleå*
*Sweden*
*Phone: +46-920493549*
*Mobile: +46-730821280*
*e-mail: joel.bh.andersson@ltu.se*

**Response to the main concerns:**

[Figure]
 1. **Comment from referee:**

The wording of the manuscript is poor. The authors are imprecise and don't use the language of our science strictly. They mix up terms and use language that makes understanding of their descriptions difficult or impossible. The entire manuscript needs careful rewording and possibly the care of a native speaker."

**Authors response:**

The linguistic and terminology suggestions made by JK in the attached document "se-2019-150-RC2-supplement" significantly improved the manuscript and we will continue to work on the language as suggested in the detailed comments below. Changes already made to the manuscript are found as answer to comments in the attached document "se-2019-150-RC2-supplement". Co-author Edward Lynch (EL) is a native English speaker and he has been involved in the full writing process. EL will carefully assess the manuscript again from an English language and wording perspective before resubmission.

We are also continuing to work on the manuscript to make sure that geological terms are used correctly and consistently in accordance with the detailed comments below. However,

already at this early stage of reworking the manuscript, we note that some of the terms criticized by JK are indeed used correctly (see below).

**Author's changes in manuscript**

The majority of the suggested changes in the supplementary material "se-2019-150-RC2-supplement" attached by JK have been accepted and corrected in the manuscript. Regarding the usage of some terminology we disagree with the review. As pointed out by JK in the detailed comments, the manuscript does contains inconsistent use of some terms, and this will be corrected in the resubmitted manuscript. However, we regard that some terms criticized in the supplementary material (se-2019-150-RC2-supplement) but not discussed further in the detailed comments should not be changed. These suggested changes of terminology include:

*"Deformation phase" suggested to be changed to "deformation stage". The argument by JK is that the word or term* **phase** *"is a term that in our science general means mineral or molecule."*

**Motivation:** The term "deformation phase" is commonly used in structural geological literature. In the manuscript, we use the term "deformation phase" according to the definition by Fossen (2010): "A deformation phase is a time period during which structures formed continuously within a region, with a common expression that can be linked to a particular stress or strain field or kinematic pattern".

*"Deformation event" suggested to be changed to "deformation stage". The argument by JK is that an "event" is very short lived, e.g. an earthquake.*

**Motivation:** This is a good point and we agree that the term "event" intuitively gives the impression of a geologically instant happening. However, neither "phase" or "event" implies a length of duration of a particular geological process. The term "deformation event" as well as "metamorphic event" is used frequently in our science and are used in scientific journal papers as well as books. See for example the paper by Kärki & Laajoki (1995) in the Journal of Structural Geology where both the term "deformation event" as well as "deformation phase" are used interchangeably. Vernon & Clarke (2008) also use the usage of "metamorphic event" to generally describe the occurrence of a stage of metamorphic process or change in space and time, but without any specific absolute constraints on its duration.

*"Mineral association" suggested to be changed to "mineral assemblage".*

**Comment:** We apologize for the inconsistent use of the terms "mineral association" and "mineral assemblage". Both terms are used in the revised manuscript. However, in the resubmitted manuscript, the term "mineral association" is the preferred term we wish to use throughout the paper.

**Motivation:** We have chosen to use the term "mineral association" instead of "mineral assemblage" in this manuscript because the term "mineral assemblage" implies equilibrium between the minerals assigned to a given assemblage. Many of the mineral associations described in this manuscript may be true mineral assemblages, but confirmation of this would require further studies beyond the scope of our mapping study (e.g. microprobe analysis and PTX modelling). We consider such a study an important "next step" in the geological research of the Western Supracrustal Belt (WSB) and such a study would justify a journal publication on its own.

We consider the definition by Pirajno (2009) as suitable in this context and he makes the following distinction between "mineral assemblage" and "mineral association": "A mineral assemblage refers to a group of minerals that formed more or less at the same time and are stable together. A mineral assemblage essentially defines the physico-chemical conditions of the system. A mineral association, on the other hand, is a group of minerals that occurs together, but are not necessarily in equilibrium and did not form at the same time. "

[Figure]
 *Change "movement" to "deformation".*

**Motivation:** We prefer to keep the use of "movement" in all cases where the kinematics of a structure is in focus. Our argument is that "deformation" by its nature can be "brittle" or "ductile" or a combination of both, but cannot be assigned the adjectives "sinistral", "dextral" or "reverse", which describes the relative movement of e. g. a hanging wall block of a normal fault with respect to the footwall block.

[Figure]
 **2. Comment from referee:**

The authors need to reword the entire manuscript and need to follow the two principles of writing a geology manuscript: (1) old structures or rocks need to be described before their younger counterparts; and (2) data needs to be presented and described first, interpretation follows.

**Authors response:**

Already after working through the document according to the supplementary material "se-2019-150-RC2-supplement", the manuscript has improved in line with this comment. We expect further improvements on this point in the resubmitted manuscript when the detailed comments below have been taken into account.

**Author's changes in manuscript:**

We will change in accordance with the referee comment.

[Figure]
 **3. Comment from referee:**

The metamorphic and hydrothermal assemblages need to be described in much more detail. What is their relationship to foliations and lineations? The mineralogy of hydrothermal alteration zones depends on P, T, X and physicochemical parameter. This results in the situation that hydrothermal mineral assemblages may not only vary on relative timing in the geological evolution and along a PT path. They also vary with host rock composition, fluid composition, distance from the main fluid conduit etc. This causes in many situations complex hydrothermal alteration patterns and zoning in hydrothermal ore deposits. This is well-described in many similar systems elsewhere in the world. The authors need to be more careful with their petrological data and must observe and interpret with much more detail. It would help the reader, if the authors could add hydrothermal alteration zones to their lithological and structural maps.

**Authors response:**

With this comment, JK makes precise and valid points highlighting the complexity of studying hydrothermal alteration systems. One of our aims in the paper was to keep a descriptive or

qualitative tone and try not to over interpret our field observations, particularly regarding the type, style and distribution of hydrothermal alteration.

We have discussed several times whether we should try to produce alteration maps for the study area. During our work we realized that such maps would fit into the overall aim of our study but that spatial distribution uncertainties would be extremely high since the amount of outcrop in the WSB is approximately 1% of the surface area (typical for Norrbotten). Nevertheless, after receiving this comment on including alteration on maps we agree with JK and will go forward with a regional-level alteration map for the study area.

**Author's changes in manuscript:**

We will include a figure showing a belt-scale alteration map. It will be based on mapping observations from the Geological Survey of Sweden (of which the 3$^{rd}$ author has directly contributed), combined with our own field observations from this work. The map will contain high uncertainties and we will try to communicate these uncertainties using appropriate symbols and colours, and a qualitative/descriptive approach in order to be as clear as possible.

**Response to the detailed comments:**

[Figure]

1. **Comment from referee:**

   Palaeoproterozoic: The stratigraphic commission has changed the general way of spelling this into Paleoproterozoic (also Archean, etc.)

   **Authors response:**

   Thank you for pointing this out.

   **Author's changes in manuscript:**

   It will be changed throughout the manuscript.

[Figure]

2. **Comment from referee:**

   Sulphide: The now generally accepted spelling of this word in economic geology papers is "sulfide".

   **Authors response:**

   In the manuscript we have kept the language to British English.

   **Author's changes in manuscript:**

   "Sulphide" will be changed to "sulfide" in the resubmitted manuscript.

[Figure]

3. **Comment from referee:**

   Title: Delete "meta" and use supracrustal belt. I personally prefer "greenstonebelt", because often not all of the rocks contained in such a belt are strictly supracrustal (you also describe dykes for example).

   **Authors response:**

We agree on that the prefix "meta" is not needed in the title. We also consider it as not needed throughout the manuscript since all rocks in this study are metamorphosed to some degree.

The composition of the volcanic rocks in northern Norrbotten do bear some similarities to greenstone belts elsewhere and based on chemical composition alone, it is hard to resolve the tectonic setting (e. g. read back-arc vs. mantle plume). However, the volume of basalts are subordinate in comparison to andesite and felsic compositions, which makes an important difference to the Rhyacian "true" greenstone belts that are found at stratigraphically lower positions in Norrbotten (see Bergman et al. 2001, Bergman 2018). Our main objection towards the use of "greenstone belt" for the Orosirian (c. 1.9 Ga) supracrustal belts is that it would introduce confusion into the Norrbotten geological literature. The term "greenstone belt"or "greenstones" in a Norrbotten/north Fennoscandia context has never been used to describe Orosirian rocks despite over 100 years of geological research in the area. Instead, this term is restricted to Rhyacian (c. 2.1 Ga) rift-related mafic rocks that forms part of a large igneous province stretching from northern Norway to Russia. However, we would encourage a discussion of the use of the term "greenstone belt" in Norrbotten but since we are not convinced that that "greenstone belt" is the best term we don't want to initiate the terminology in this study.

A minor but important objection towards the use of "greenstone belt" in this context is that the term "greenstone belt" is a rather loosely defined term and would require a detailed definition if used in this study. See Lynch et al. (2018) for a short summary regarding use of names and terms for greenstone-type rocks in northern Norrbotten.

**Author's changes in manuscript:**

We will exclude the prefix "meta" throughout the manuscript and motivate this early in the text.

We prefer to keep the term "supracrustal" throughout the manuscript.

[Figure]

4.  **Comment from referee:**

Mineralization: "Mineralization" is a process not a thing. Check your wording accordingly.

**Authors response:**

We agree on that mineralization is not a thing.

**Author's changes in manuscript:**

We will reword all passages in the manuscript where the term "mineralization" is described as a thing.

[Figure]

**Comment from referee:**

Introduction: There is a lot, which is repeated and detailed in later chapters. The introduction should introduce the problem and specify the research questions and the approach. This is only partly true here. Why is the study important? What will the addition to science be? Why is relating hydrothermal alteration to structures important? How is the situation elsewhere in similar terranes with IOCG deposits, Canada, Brazil, Australia (e.g. Tennant Creek, Mount Isa), Mauritania. . .. . ..?

**Authors response:**

We have reviewed the introduction and we do not find the chapter too bad. However, the points highlighted by JK in the referee comment is very important and we think that the manuscript will improve by address these question in a clearer and more concise way.

**Author's changes in manuscript:**

We have changed the chapter in accordance with the supplementary material "se-2019-150-RC2-supplement". We are currently working on improving the introduction further by addressing the questions raised in this referee comment.

[Figure]

5. **Comment from referee:**

Regional Geology: This chapter is poorly worded and poorly structured. The data and observation must be presented before interpretation. A clear stratigraphy is necessary. I suggest preparing a table or a sketch to help the reader. In the text, the nomenclature has to be used strictly and consistently.

**Authors response:**

We agree that a figure is needed showing the stratigraphy of northern Norrbotten.

**Author's changes in manuscript:**

We will reword and try to restructure the chapter as well as make sure that nomenclature is used consistently. A figure will be produced explaining the stratigraphy of northern Norrbotten.

[Figure]

6. **Comment from referee:**

Methods: No Leapfrog model is shown in this paper – adjust this chapter to the methods used for generation of the data presented in this manuscript.

**Authors response:**

We apologize for this. Leapfrog was used initially in order to group and plot data as well as to produce initial stereographic analyses. However, this work is not presented in the manuscript and we agree that it should be deleted.

**Author's changes in manuscript:**

We will delete the text mentioning Leapfrog.

[Figure]

7. **Comment from referee:**

Results: Describe your data from old to young. This is a geology paradigm that makes sense, because old structures are always overprinted by young structures. Don't shift between scales. Make a description at regional, district, local, outcrop, sample, thin section scale and organize accordingly.

**Authors response:**

We agree that some passages in the text require restructuring (old-young) and that scales, especially time scales (e. g. Ma and Ga) are mixed in the text.  Many such passages have been pointed out in the supplementary material "se-2019-150-RC2-supplement".

**Author's changes in manuscript:**

We have made the changes suggested in the supplementary material "se-2019-150-RC2-supplement" on this comment.

[Figure]

**8. Comment from referee:**

Chapter 6.4.1 is needed much earlier in the manuscript.

**Authors response:**

Chapter 6.4.1 is a summary section, which is usually given at the end of a paper.

**Author's changes in manuscript:**

During the reworking of the manuscript, we will try to find a way to introduce a summary of hydrothermal alteration, metamorphism, and its relation to deformation earlier in the manuscript. For example, a summary paragraph could be included at the end of each relevant section to remind the reader of what was just presented and its significance. This should help the reader to stay with the overall narrative of the paper.

[Figure]

**9. Comment from referee:**

Figures: Correct legend and check for completeness of all legends in all maps.

**Authors response:**

We note the problem with legends.

**Author's changes in manuscript:**

We will change the legends accordingly.

[Figure]

**10. Comment from referee:**

Add legend to the stereographic plots.

**Authors response:**

The stereoplots do have legends inside the plot circle. We prefer to keep it this way because it saves a lot of space on the page.

**Author's changes in manuscript:**

We prefer to keep it this way because it saves a lot of space on the page and makes the stereo plot easier to read.

[Figure]

**11. Comment from referee:**

Page 1, Line 16 ff.: Relationship between alteration assemblages is unclear. Why are there two regional hydrothermal alteration assemblages? What is their relationship? What is the importance of calcite? This needs explanation that is more careful and rewording.

**Authors response:**

According to the descriptive approach we have chosen in this manuscript, these mineral associations are to be considered as separate because they show overprinting relations to each other. However, we realize that these mineral associations may form part of an evolving

hydrothermal system, which we state on page 12 line 10. The benefit of our approach is that it provides a scheme that is relatively easy to apply at the outcrop scale and does not lead to overinterpretation of how the hydrothermal systems evolved through time-space. We think this approach may be of some benefit to future geoscientists working in this area and also exploration geologists.

The relationship between the magnetite + amphibole and albite + scapolite alteration is that they are broadly coeval because they show crosscutting relationships to each other, which is stated on page 1, line 16.

The role of calcite is a very legitimate and good question and we thank JK for highlighting this. The simple answer is that we are uncertain about its role during either D1 or D2. However, calcite is present and we consider it important to describe in order to give a full view of the alteration minerals we have observed. One possibility, similar to that described by Kesler (2005), is that calcite formed at a relatively late stage of D1 and D2 during ingress of late fluid(s), facilitating the retrograde (down-welling) deposition of carbonate. The ultimate source of this carbonate would likely have been the supracrustal rocks which would have experienced some degree of decarbonation-devolatilization during regional metamorphism.

**Author's changes in manuscript:**

We will try to clarify the questions raised in this referee comment by explanation that is more concise and rewording of the relevant section.

[Figure]

**12. Comment from referee:**

Page 1, Line 20: Avoid the term "brittle-ductile". This term is derived from geophysical (seismic) investigation of the Earth's crust and defined as a zone of velocity change of seismic waves. It has no geological meaning. I can show you examples of brittle-ductile behaviour of rocks at 250∘C and at 650∘C. I suggest avoiding the term, because it does not add any information.

**Authors response:**

This is a good point. The term brittle-ductile is often used within structural geology to describe structures that contain both brittle and ductile components formed at the same time. This is also what we aim to describe by the term. As pointed out by JK, depending on many factors, such as mineralogy, strain rate, access to fluids etc., structures showing both brittle and ductile components form over a large temperature span. However, we believe use of this term adds geological information by emphasizing that a particular structure may show both brittle and ductile characteristics.

A few alternative terms could also be considered. The term "low grade" is one alternative frequently used. However, that term does not describe whether a structure contains only brittle- or only ductile-type deformation, or a mix of brittle and ductile structures.

Fossen (2010) defines plastic deformation as: ..."the permanent change in shape or size of a body without fracture, produced by a sustained stress beyond the elastic limit of the material due to dislocation movement." The term "brittle-plastic deformation" may here be the better term to use as it describes that crystal plastic deformation occurred as well as brittle

deformation during the same deformation event (or stage). However, this term would still not bridge the dilemma on temperature since e. g. quartz and feldspar deforms by crystal-plastic deformation at different temperatures. The benefit of describing this from the WSB is that the presence of brittle components in the shear zones is a simple distinction between D1 and D2 structures. Brittle components have not been observed in structures that we interpret as D1-related whereas, brittle components have been observed together with ductile components in D2-structures.

**Author's changes in manuscript:**

We will change the term brittle-ductile to brittle-plastic in order to keep the description of brittle and ductile components formed together. We hope that our argumentation is valid on this point.

[Figure]

**13. Comment from referee:**

Page 2, Line 1: This sentence is misleading as is the referencing. The two papers do not describe the situation in Norbotten as your wording indicates. It is unclear to me, how metamorphic rocks with only restricted porosity (and generally no permeability) can focus fluid flow. Fluid follows the hydraulic gradient and can only migrate through permeable rock.

**Authors response:**

We agree that the references do not describe the situation in Norrbotten but addresses deformation and fluid flow from a fundamental point of view.

**Author's changes in manuscript:**

Text will be deleted in the resubmitted manuscript.

**14. Comment from referee:**

Page 2, Line 3 ff.: This is repeated from the previous page. Reword and avoid repetition.

**Authors response:**

We cannot find the repetition.

**Author's changes in manuscript:**

During the further reworking of the document we will see if we can locate what is addressed in this referee comment and reword in order to avoid repetition.

[Figure]

**15. Comment from referee:**

Page 2, Line 12: There is a problem here and also elsewhere in the text with the terminology. You define "Svekokarelian" as an orogeny. Here you use it as a stratigraphic term. This is confusing for the reader and needs to be avoided. In the regional geology chapter, a description of stratigraphy (old to young) and metamorphism and deformation (if information is available) is needed. The terminology needs to be clearly defined (if helpful with a table) and strictly used in the entire manuscript.

**Authors response:**

We agree that a figure explaining the stratigraphy of Norrbotten is needed for the manuscript. Such a figure will help clarify the confusion addressed by this referee comment.

However, even though the understanding of the timing of metamorphism and deformation has increased during the last 20 years of research, it is still relatively poorly understood. Nevertheless, including deformation and metamorphism in such a figure would be a beneficial summary of the geology in northern Norrbotten.

**Author's changes in manuscript:**

For the resubmitted manuscript, we will insert a figure explaining regional stratigraphy, metamorphism and deformation for Norrbotten.

[Figure]

**16. Comment from referee:**

Page 4, Line 25 ff.: What was used to constrain the PT conditions? The method or mineral assemblage needs to be stated. Otherwise, the reader cannot evaluate the quality of the data himself. Is it the metamorphic peak that is recorded? What is the approximate timing of metamorphism? You state that PT conditions raised from greenshist to amphibolite facies regionally, but then outline granulite facies conditions. What is true?

**Authors response:**

Bergman et al. (2001) does not go into to the details of this. The PT-conditions were based on mineral associations typical for the metamorphic zones. The mineral associations were supported by mineral chemistry obtained by microprobe. Most data comes from the high-grade areas.

The metamorphic evolution in northern Norrbotten is poorly constrained. Reset U-Pb ages indicate that there were at least two metamorphic peaks, M1 at approx. 1.9 Ga and M2 at approx. 1.8 Ga. However, the role of contact metamorphism etc. is an open question.

Granulite metamorphic facies has been recorded in volcanic rocks near 1.8 granites. This data probably reflects the influence of contact metamorphism.

**Author's changes in manuscript:**

"limited PT-modelling" has been changed to "microprobe data".

The role and distribution of granulite facies metamorphism will be clarified in the text.

[Figure]

**17. Comment from referee:**

Page 4, Line 30 ff.: You say that the deformation is polyphase, but you only describe one stage of deformation in the text.

**Authors response:**

The text summarizes the literature that outlines at least two deformation events in N Norrbotten.

**Author's changes in manuscript:**

The paragraph will be reorganized and clarified in order to highlight D2 in a clearer way.

[Figure]

**18. Comment from referee:**

Page 10, Line 6: This needs much more explanation. How does dip-slip relate to compressional deformation? Generally, compressional deformation is related to folding, strike-slip and reverse shearing and not normal (dip-slip) deformation.

**Authors response:**

This referee comment points out the missing word "reverse" before "dip-slip" in this paragraph. We thank JK for pointing out this typing error. After adding "reverse" to the context, the sentence states that the relative movement of the shearing is "reverse dip-slip" that is the same as reverse shearing with the additional information that the movement occurred along the dip-direction of the plane.

**Author's changes in manuscript:**

This referee comment have been corrected for in accordance with the comment in the supplementary material "se-2019-150-RC2-supplement".

We will check through the document in order to make sure that "dip-slip" is always stated "reverse dip-slip" since we have not observed structures showing normal dip-slip kinematics in this study.

[Figure]

**19. Comment from referee:**

Page 10, Line 10 ff.: You need to organize. E.g. you describe brittle and ductile deformation of feldspar, then describe many other things and then describe ductile deformation of quartz. See above; organize the structural description according to scale, either from small-scale to regional-scale or vice versa. The quartz fabrics (D1/D2 similar, low-T) contradict the feldspar fabrics (D1 >450∘C, D2 low-T) – why?

**Authors response:**

We agree that this passage in the text would benefit from a reorganization.

The question addressed by JK on the quartz textures is a very good question that we do not have a good explanation for at the moment. We will address this in the resubmitted manuscript based on our interpretation and literature context.

**Author's changes in manuscript:**

We will try to reorganize the text passage and try to address the contradiction of the quartz textures.

[Figure]

**20. Comment from referee:**

Page 12, Line 15 ff.: This remains unclear. If scapolite forms porphyroblasts, then it at least postdates D1 (otherwise, it must be porphyroclastic). If the veins are deformed by D2, then they predate D2. Which time constraints do you have between D1 and D2 to say that the veins postdate the scapolite porphyroblasts? Why is scapolite not formed at the same time?

**Authors response:**

The temporal relationship between porhyroblasts and deformation is described in detail by e. g. Passchier & Truow (2005). Porhyroblasts can form as pre- syn- or post-tectonism. Unfortunately, our thin section material from the WSB do not contain any scapolite porphyroblasts and thus we cannot show the timing of the scapolite porphyroblasts other

than using our outcrop observations. However, we have studied scapolite porphyroblasts relative to deformation in thin section from the Pahtovaare- and Nunasvaara- areas east of the WSB. In both these places, the scapolite porphyroblasts show a syn- to post-timing (syn-D1 in Pahtovaare and syn-post-D1 in Nunasvaara) relative D1 and predate D2. However, because scapolite porhyroblasts constitute "rigid bodies" surrounded by weaker minerals they are not noticeably deformed by D2-deformation. We can provide images if there be of interest.

We agree that veins that are deformed by D2 must predate (or at least have been formed during) D2. We will carefully review and rework this passage in the text to make sure that our interpretations are consistent with our observations.

**Author's changes in manuscript:**

The timing of scapolite will be clarified in the text.

[Figure]

**21. Comment from referee:**

Page 13, Line 4 ff.: This is very confusing. You jump in your description from early to young and vice versa. The reader is unable to follow this. The important message is, which hydrothermal alteration assemblage(!) is temporally related to which structural fabric and to which stage in the metamorphic PT evolution (prograde, peak, retrograde). You also need to distinguish between hydrothermal alteration and metamorphic mineral assemblages. They form via completely different processes. This needs careful rewording.

**Authors response:**

We agree that this paragraph describing epidote in different mineral associations is confusing and could be more clearly structured.

**Author's changes in manuscript:**

We will rearrange this paragraph, or possibly delete it and move parts of this text to other paragraphs describing mineral associations where epidote is a component mineral.

[Figure]

**22. Comment from referee:**

Page 14, Line 20 ff.: What about the feldspar microstructures described above? What do they indicate? What about the mineral assemblages? Which PT conditions do they indicate? This needs much more discussion and integration of microstructural and petrological data. Likely much better PT constrains are possible by such an integration.

**Authors response:**

This is a good point.

The feldspar microstructures indicate that the temperatures were lower during D2 in respect to D1 because brittle feldspar is only observed in E-W-trending D2-structures.

**Author's changes in manuscript:**

We will try to develop this in the resubmitted manuscript.

[Figure]

**23. Comment from referee:**

Page 15, Line 5: Relatively low P and upper crustal conditions are also true for D1 deformation. This does not add any information and is a very vague and imprecise statement. See also comment above.

**Authors response:**

It is true that both D1 and D2 are rather low grade, however, observations of D2 structures indicate that the deformation took place higher up in the crust in respect to D1. We believe that this is rather well described for D2 but the manuscript probably requires a better description for D1 and what is indicated in these D1-structures with respect to inferred P-T conditions.

**Author's changes in manuscript:**

We will try to address an estimation about inferred P-T conditions for D1-structures.

[Figure]

24. **Comment from referee:**

Page 15, Line 6 ff.: This needs more discussion. It remains completely unclear, why you add information from other localities and which relevance that may have. You first need to constrain PT conditions and relative timing of deformation, metamorphic and hydrothermal stages in your study area before you can compare those to other (similar or connected) areas. Readers, who are not familiar with the regional geology, will be confused and cannot decide on the relevance of the data presented here for comparison.

**Authors response:**

The relevance of comparing results from nearby areas to our observations in the WSB is that it is important to highlight what observations agree with the current understanding of the northern Norrbotten area. We think that such comparisons are important also from areas that are not fully understood (e. g. the WSB). We do not consider this study will be the last on the WSB, but we rather hope it represents an initial study to build further upon by future work.

**Author's changes in manuscript:**

We will clarify in the text the importance of comparing the results in this study to results from adjacent areas and place an emphasis on our results and interpretations and how they then may relate to other localities (if needed).

[Figure]

25. **Comment from referee:**

Page 15, Line 13 ff.: This part suffers from bad wording and poor organization. Please reword and discuss from old (L1) to young (L2). The mineral species that define the stretching lineation may help in constraining relative timing and PT conditions.

**Authors response:**

The beginning of the paragraph is described from old (L1) to young (L2). The end of the paragraph discusses why L1 and L2 can be separated and that they are best explained by different shortening directions. We agree that the end of the paragraph may be improved by reorganization and rewording or by making two paragraphs of this long paragraph.

**Author's changes in manuscript:**

The paragraph on page 15 including line 13 will be carefully reviewed and reorganized in the resubmitted manuscript.

[Figure]

**26. Comment from referee:**

Page 16, Line 20 ff.: Why are you presenting this? There is a lot of speculation and the reader gets no idea, what the relevance of this is: delete.

**Authors response:**

The reason why this discussion was presented in the manuscript is that there is a paradox in the interpretations on the tectonic setting for the earliest deformation history in northern Norrbotten. Studies based on structural geology tend to argue for a compressional setting and the development of a fold-and-thrust belt. In contrast, studies based on petrology tend to argue for an extensional setting (back-arc or mantle plume). We think that it is relevant to compare structural results to petrological results in order to discuss the overall tectonic setting in the area. But it is possible that we can do that in a much clearer way or in other parts of the manuscript.

**Author's changes in manuscript:**

We will clarify this, either, by deleting this text or by moving parts of the text to other paragraphs.

[Figure]

**27. Comment from referee:**

Page 16, Line 26 ff.: I am confused: Why do D2 shear bands indicate D1 deformation?

**Authors response:**

Because there is an overprinting relationship. Due to the presence of D2-shearbands that deform the S1-fabric, we interpret the S1-fabric to be D1-related and the overprinting shear band as a D2-related structure.

**Author's changes in manuscript:**

We will try to clarify this in the resubmitted manuscript.

[Figure]

**28. Comment from referee:**

Page 17, Line 2 ff.: What is the timing of metamorphism? The epidote-amphibolite facies in pillow basalts can be very early in the evolution and may represent seafloor metasomatism. Alternatively, a similar mineral assemblage can form during regional metamorphism. This needs more discussion of evidence of relative timing.

**Authors response:**

This is a valid question. We will present evidence for syn-D1 growth of hornblende in the same sample as shown in Figure 15 G in order to clarify the timing of M1 (this will not be Fig. 15G anymore after inserting the figure on the stratigraphy in northern Norrbotten in accordance with earlier referee comments).

**Author's changes in manuscript:**

We will add a figure in fig. 15. We may be forced to delete one image from Fig. 15 in order to make room for this new micrograph. We will take a discussion about this and possibly consult the editor in how we solve this issue.

[Figure]

**29. Comment from referee:**

Page 19, Line 4: If the scapolite is really porphyroblastic everywhere, then it must (by definition) be post-tectonic. You say that the scapolite porphyroblasts occur in low-strain areas, which may mean that scapolite growth postdates D1 here but is pre-, syn- or post-D1 in the high-strain zones. This needs more discussion and more precise investigation of the relative timing. Furthermore, you contradict yourself with the interpretation of the absolute age data. You say that D1 is 1.88-1.86 Ga, that scapolite alteration is syn- to late-D1 and that scapolite formed (together with titanite) at 1.9 Ga. This is 20-40 m.y. earlier than D1 and not syn- to late-D1. Moreover, you do not provide the precision of the geochronological data, which makes further evaluation by the reader impossible. Is scapolite formed during seafloor alteration at 1.9 Ga?

**Authors response:**

For the comment on scapolite porhyroblasts, see our response on the referee comment regarding Page 12, Line 15 ff. We agree on that that the timing of scapolite porhyroblasts can be clarified further in the text.

The age obtained by Smith et al. (2009) is reported at 1903±8 Ma.

JK points out important weaknesses in the understanding of the early scapolitization in northern Norrbotten. This question deserves attention and his comment highlights the need for further geochronological data on scapolite-altered rocks in Norrbotten. Scapolite has never been studied from a structural geology perspective in Norrbotten, hence, we hope that our field observations can add information to the overall picture.

**Author's changes in manuscript:**

We will reconsider to draw any parallels between scapolite porhyroblasts in the WSB and the titanite age of a scapolite-altered mineral association reported by Smith et al. (2009).

[Figure]

**30. Comment from referee:**

Page 19, Line 10 ff.: The discussion about Cl/Br geochemistry and fluid source remains unclear. If you have two generations of scapolite, what will whole rock data tell you? Why do you consider an evaporitic source, when the data contradicts this? This needs much more discussion, if this is important for the conclusions of this work. I am not sure, where this will lead to?? You don't really have shown constrains on the relative timing of scapolite alteration. This makes it really difficult to follow.

**Authors response:**

We have reworded this passage while working through the comments in the supplementary material "se-2019-150-RC2-supplement". However, we find this text targeted by this referee comment as slightly out of scope for this study and the text is probably not needed. Thank you for pointing this out.

**Author's changes in manuscript:**

This text will be deleted in the resubmitted manuscript.

[Figure]

**31. Comment from referee:**

Page 19, Line 17 ff.: If you would follow the geological principle of describing the oldest features first, your text would be much easier to follow. You get not all possible information out of your data. There is much more on P-T-D-X relationships in your data that needs to be presented and discussed. The important part is that hydrothermal alteration assemblages are controlled by complex P-T-X relationships. The mineral presence in the hydrothermal alteration assemblages depends not only on fluid composition, but also on host rock composition and fluid rock ratio. Thus, it is very common in hydrothermally altered regions to find different hydrothermal alteration assemblages in different host rocks that formed at the same time or to find zoning of hydrothermal alteration assemblages depending on the distance from the main hydrothermal fluid conduit (e.g. a shear zone or a pluton). All this needs to be worked out from your data and displayed in maps and discussed accordingly.

**Authors response:**

We agree with JK that there is probably much more to the P-T-D-X relationships in WSB than is shown by this study. We agree that this constitutes a very interesting question and deserves further attention from the research community. However, we consider a thorough thermodynamic investigation of the mineral alteration systems in the WSB as the next step to take in the geological research in the area. Such a study would require a different approach than what we have chosen in this study. In this study, we present geological mapping results performed during three field seasons (May-October) which forms a part of the lead authors ongoing PhD research (see Andersson et al. 2017, Andersson 2019, Andersson et al. 2019). Our observations are backed up by a petrographical investigation focused on kinematics of the dominant structures and linked to the mineral alteration associations that we have observed throughout the belt. We find the suggested work to be beyond scope for the current study but we encourage further work on the complex questions addressed by this referee comment.

**Author's changes in manuscript:**

As mentioned in comments to previous referee comments, we will make sure that our descriptions and/or discussions follow the overall structure old-to-young. We agree on that this makes sense. However, in some cases different generations may need to be alternatively described in relation to each other in the same sentence in order to make the context as clear as possible.

In the resubmitted manuscript, a mineral alteration map will be included. However, it is important to note that this map will carry uncertainties that must be considered as high. We will try to communicate this as clear as possible in the resubmitted manuscript.

[revised manuscript text omitted]